# A sixfold urban design framework to assess climate resilience: Generative transformation in Negril, Jamaica

Tapan Kumar Dhar[1]*, Luna Khirfan[2]

**1** School of Urban Planning, McGill University, Montreal, QC, Canada, **2** School of Planning, University of Waterloo, Waterloo, ON, Canada

* tapan.dhar@mcgill.ca

## Abstract

The uncertainty of climate change's impacts hinders adaptation actions, particularly micro-scale urban design interventions. This paper proposes a sixfold urban design framework to assess and enhance the resilience of urban form to climate change, where urban form refers to the patterns of streets, buildings, and land uses. The framework is then applied to Long Bay in Negril, Jamaica–a coastal area that incorporates the complex interactions between urbanization and a highly vulnerable socio-ecological system to climate change-related hazards, primarily sea-level rise. Empirical evidence from 19 in-depth interviews with planning and design professionals and development actors, in situ observations, and morphological analyses reveal that Long Bay's current adaptation strategies heavily rely on bounce-back resilience measures that predominantly consider the impacts of extreme climatic events rather than slow-onset ones. Such strategies abet current tourism-driven development patterns while overlooking Long Bay's inherent abilities for generative transformation and incremental changes to meet climatic uncertainty. Instead, this study's findings highlight how generative urban form transformation would better equip Long Bay to cope with future uncertainty–climatic or other.

## 1. Introduction

> *Most of Jamaica's social and physical infrastructure is in coastal areas. We can't move them. Thus, we need to adopt coastal defence mechanisms such as hard structures-break waters, sea-walls and so on to protect them.*

The statement illustrates how a professional planner hints at the challenge of rising sea levels in Jamaica and the adaptation preferences. Many small island developing states (SIDS) world-wide heavily depend on their coastal resources and experience similar challenges, hence, seek sustainable adaptation to climate change. taking the right decision for sustainable adaptation planning is crucial, since we are living in the age of climatic uncertainty. Are such coastal defences good enough to protect people and settlements? If so how long can they adapt to the climatic impacts that we are not fully aware of? How does the planning and design of coastal settlements be more resilient with less impacts on socio-ecological system? Many questions as such are around us.

**Data Availability Statement:** All relevant data are available in Supporting Information files.

**Funding:** The Partnership for Canada-Caribbean Climate Change Adaptation (ParCA) provided financial support for this study. The grant number

is 106372-006. The funders had no role in study design, data collection and analysis, decision to publish, or preparation of the manuscript. No authors received a salary from the funder (i.e., ParCA). We received funding from ParCA.

**Competing interests:** The authors have declared that no competing interests exist.

In addressing these questions, over the past decade, scholars have highlighted the lack of theoretical and empirical connections between resilience and climate change adaptation and their applications to urban design [1–6]. In this paper, urban design refers to a distinct field, recognized since the 1950s The first Urban Design Conference at Harvard University in 1956 was the beginning of urban design as an intellectual discipline and as a professional field distinct from urban planning, architecture, and landscape architecture. Since then, urban design has highlighted three major components: it conserves the natural environment; it creates a desirable public realm (civic design) that includes transportation, streets, civic spaces, shopping, entertainment, parks, and recreation; and it supports social interaction in residential neighborhoods, workplaces, and mixed-use downtowns [7, p.2], which associates with architecture, city planning and landscape architecture and coordinates different elements of a city in a three-dimensional sense [8, 9]. The connections are crucial for enabling human settlements to adapt to climate change through t configuring urban design elements, such as street networks, buildings, and land uses. This paper aims to transfer resilience, a concept that is rooted in the socio-ecological literature (embedded in the natural sciences), to urban design and planning and landscape architecture (grounded in the social sciences and the humanities, respectively). We, in line with Olsson et al.'s [10] recommendations, avoid deploying resilience "as a grand or unifying theory" and instead explore its compatibility with "some, [but] not all" (p.9) urban design concepts. Accordingly, in this study, urban resilience becomes a "guiding principle" as opposed to "an end state" [11] (p.17). It encompasses a range of actions that enhance a system's capacity to recover from past disasters (ex-ante) as well as those that improve its ability to cope with future uncertainty (post-ante) [1, 4].

Since the last decade, a plethora of planning literature has highlighted the different theoretical discourses on resilience vis-à-vis climate adaptation and its applications–albeit mostly at the regional scale [12]. In general, both resilience and adaptive capacity aim to reduce vulnerability and minimize, if possible, exposure. However, the climate change and the socio-ecological literature define many concepts, like resilience, adaptation, and vulnerability, differently. This definitional and relational ambiguity often hinders understanding their theoretical and operational connections to urban design [13–16]. As Lawrence Vale [17] notes, "most leaders have so far done little to adapt their cities, or to acknowledge ecological limits and ongoing vulnerability when building or rebuilding" (p.828). Therefore, the adaptation and resilience applications, whether proactive, anticipatory, or generative (the latter is the focus of this paper), remain limited in the field of urban design. Masnavi et al. [18] identify two major lacks in urban resilience studies: connecting spatial morphology to resilience systems and understanding the spatial form of a city to support urban socio-ecological systems. They both fall within the realm of urban design and are different from the "social-science-predicated policy approach" common in the field of urban planning [19]. Urban design comprises six dimensions: morphological, perceptual, social, visual, functional, and temporal [20]. This paper's focus is on urban morphology Urban form and morphology are widely used in the field of urban design. In general, urban form refers to "the physical environment" of a city that includes the spatial pattern of its permanent and inert physical objects such as hills, rivers, streets, buildings, utilities, and trees [21]. In particular, it encompasses the unique morphological characteristics of a town, i.e., its "physiognomy or townscape," which combines the town plan, the pattern of building forms, and the pattern of land use [22]. The town plan itself includes street networks, blocks, and building footprints. Urban morphology, a study of urban form, highlights the physical dimension of urban design, which refers to the "physiognomy, townscape" or urban form Urban form and morphology are widely used in the field of urban design. In general, urban form refers to "the physical environment" of a city that includes the spatial pattern of its permanent and inert physical objects such as hills, rivers, streets,

buildings, utilities, and trees [21]. In particular, it encompasses the unique morphological characteristics of a town, i.e., its "physiognomy or townscape," which combines the town plan, the pattern of building forms, and the pattern of land use [22]. The town plan itself includes street networks, blocks, and building footprints. Urban morphology, a study of urban form, highlights the physical dimension of urban design that includes the configuration of the street network, the urban blocks and parcels, and the buildings (including their three-dimensional form) as well as the pattern of land uses [22].

The research connecting urban design and resilience, whether or not related to climate change, is limited. Of this limited body of research, only a fraction integrates socio-ecological and planning theories in devising innovative urban design criteria that enhance urban resilience. Two types of such research stand out. The first establishes direct theoretical links between urban design and resilience, such as Allan et al.'s [23] eight attributes of a resilient urban environment and Sharifi's [24] properties of resilient urban streets. The second theoretically connects urban design and climate adaptation to enhance the resilience of existing urban areas and/or future urban developments, such as Feliciotti et al.'s [25] five theoretical proxies of resilient urban form and Dhar and Khirfan's [26] urban design resilience index (UDRI). Yet, both approaches are criticized for two shortcomings. Firstly, neither operationalizes their theoretical framing by means of context-specific urban design guidelines. Secondly, neither provides specific criteria for evaluating the resilience of an existing urban context. As a consequence, to date, the urban design research lacks urban morphological indicators and variables that gauge the climate resilience of the built form of human settlements. The only exceptions occur in the grey literature, such as the City of New York's *Climate Resiliency Design Guidelines* [27] and the City of Vancouver's *Resilient Vancouver Strategy* [28]. These shortcomings hinder the formulation of universal urban design concepts that may be calibrated to each context's specific needs for climate change adaptation and, by consequence, climate resilience.

Hence, this paper investigates how we may understand and assess urban resilience through the spatial layout and physical configuration of urban form. To meet this objective, this paper first addresses the inquiries and observations raised by several scholars about urban form's resilience in general and its climate resilience in particular, such as Jabareen's [5] risk city, Quigley et al.'s [29] socio-ecological resilience agendas of design, Hakim's [30] generative process of urbanism, Beatley and Newman's [31] biophilic cities, and Vale's [17] progressive resilience. This paper then investigates: 1) how can urban morphological design reduce climatic uncertainty? And, how can it enable cities to become more climate resilient, especially given the uncertainty associated with a continually changing climate?

In investigating these questions, the following section first establishes a theoretical foundation that draws upon various concepts in the urban planning and design literature that are analogous to the concept of resilience in the socio-ecological literature. Based on these theoretical connections, we identify several concepts common in both bodies of literature. The review of literature confirms that six urban design concepts are crucial to connect urban design and urban resilience. We then identify their variables that directly or indirectly impact the climate resilience of urban form, specifically, its urban morphology (i.e., street networks, buildings, and land uses). We operationalize these concepts to evaluate the resilience of Long Bay in Negril, Jamaica, a vulnerable coastal area in the Caribbean basin. Based on the empirical findings, we propose an urban morphological climate resilience framework that theoretically contributes to understanding and assessing the current resilience. This proposed framework simultaneously complements the notions of resilience in the urban design and landscape architecture discourses (the social sciences and humanities), including generative processes [30], biophilic cities [31], and progressive resilience [17]. It also connects them to socio-ecological resilience (the natural sciences). In practice, this framework informs future climate

resilience planning. Our findings also include urban design recommendations to improve the resilience of Long Bay and other similar coastal settlements.

## 2. Contemporary discourses on urban resilience

Resilience, coined by socio-ecologist Holling [32], refers to "a measure of the persistence of systems and of their ability to absorb change and disturbance and still maintain the same relationships between populations or state variables"(p.14). Similarly, urban resilience refers to the ability of urban systems "to withstand a wide array of shocks and stresses" [33] (p.164) and/or to return back from destruction whether or not such destruction is attributed to climate change [34]. Yet, resilience, in the socio-ecological literature, and adaptive capacity, in the climate change literature, both focus similarly on reducing a system's vulnerability to different disturbances and coping with uncertainty [1, 4]. Despite different disciplinary roots, resilience and adaptive capacity are used interchangeably. However, in a socio-ecological system, Carl et al. [35 p.354,375] consider resilience as "a key property of sustainability" and "a precondition for adaptive capacity" and that the loss of a system's resilience will reduce its "capacity" of dealing with changes. In other words, the capacity to cope (i.e., adaptive capacity) can be understood as components of a system's resilience [36]. Thus, in an urban context, "an adaptive system is not necessarily always resilient, but a resilient system can always be adaptive"[26 p.76].

Regardless of whether an urban system is susceptible to slow or rapid onset climatic hazards, uncertainty can be referred to "a perceived lack of knowledge" of the purpose, action, and/or outcomes [37] (p.504). Indeed, urban resilience discourse vis-à-vis climate change underscores "an increasing sense of complexity, uncertainty, and insecurity about cities and a desire to identify new adaptation and survival strategies" [11] (p.17). Accordingly, there is consensus that urban resilience needs to "focus on adaptive capacity rather than specific adaptations" since within adaptive capacity, a system adjusts to a changing climate, including climate variability and extremes [16] (p.44) (see also [38, 39]). This is because, given the complex interaction between human and urban systems, reducing only one particular climatic risk (as a protective measure) is difficult. Such disregard for the complexity of risks may inadvertently amplify other types of vulnerability across spatial and temporal scales. Scholars call this failure maladaptation, which actually exacerbates uncertainty [40]. Thus, the urban planning and design discourse is gradually transitioning from identifying "new adaptation and survival strategies" toward building resilience to enhance the adaptive capacity and address uncertainty [11]. With this in mind, Meerow et al. [16] redefine urban resilience. Firstly, their definition addresses the complexity of an 'urban system's scale whereby resilience is "the ability of an urban system and all it's constituent socio-ecological and socio-technical networks across temporal and spatial scales" (p.45). Secondly, their definition underscores urban resilience's three key objectives, which we consider to be analogous to three modes of socio-ecological resilience (i.e., engineering, ecological, and evolutionary resilience, see [1]). Meerow et al.'s [16] first resilience objective is "to maintain or rapidly return to desired functions in the face of a disturbance" (p. 45), which we compare to the engineering (bounce-back) mode of socio-ecological resilience [41] that occurs through anticipatory adaptations. Both their second and third objectives—"to adapt to change" and "to quickly transform systems that limit current or future adaptive capacity" (p.45)—highlight the attempts to cope with future uncertainty through transformative adaptations. We parallel the latter to the ecological (bounce-forward) and the evolutionary (transform-forward) resilience [26, 42]. Uncertainty is particularly salient in Davoudi et al.'s [1] definition of evolutionary resilience that includes "inherent uncertainty and discontinuities, insight into the dynamic interplay of persistence, and adaptability and transformability" (p.306). Thus, evolutionary resilience challenges previous notions of

equilibrium that exist in both engineering and ecological resilience [41]. The evolutionary form of resilience relies on the belief in multiple alternative states of a system while also favouring natural transformation over time.

The 2022 IPCC's recent assessment report also supports this evolutionary form of urban resilience because it advances 'transformation' to cope with the current and future climatic uncertainty [43]. While recent studies discuss similar theoretical trajectories and their potential connections between urban resilience and uncertainty, they focus heavily on governance and institutional capacity to manage risks rather than on urban form. For example, Ziervogel et al., [44] identify the barriers of collaborative governance to flood resilience in Cape Town, while Patterson and Huitema [45] advocate for a practice-centered approach to achieve visible changes in resilience governance. They focus on planning policies, governance, land use and spatial planning, and risk assessment [12, 46] but overlook the connections between climate uncertainty and urban design, particularly its urban morphology dimension (consisting, as mentioned earlier, of the town plan–street network, blocks/parcels, and building footprints; land-use patterns; and the built form) [22].

## 2.1. Urban form's resilience and its design attributes

Although resilience emerged in the early 1970s in the socio-ecological literature, its connection, particularly with regard to urban design and climate change, remains relatively new [1, 2, 29, 47]. According to Vale [17], urban resilience encompasses the physical built environment along with the pecuniary socio-economic and emotional-psychological components. The collective presence of all three facilitates "bouncing forward" through continuous evolution. Yet, the discrepancy between an equal emphasis on the physical built environment and the social-institutional approaches limits the research on adaptation and climate resilience [48]. To some extent, evolutionary resilience accepts a system's social psychologist's domain, as Vale [49] argued, that considers the larger interconnections of social ecology and economic well-being of an area. The numerous attributes, indicators and/or variables that assess community resilience to climate change in general focus on the socio-ecological dimensions more than the urban morphological ones, such as in the Sendai Framework (2015–30). Indeed, our extensive review of these attributes (summarized in S1 Appendix) also confirms such emphasis on the social and economic dimensions and reveals the need to operationalize the resilience of urban form. In other words, it is imperative to identify morphological determinants/attributes (i.e., urban design variables and indicators) that facilitate enhancing and assessing the resilience of urban form in the face of climatic uncertainty.

Therefore, we deployed three criteria to sort several urban design concepts that are analogous to resilience. Firstly, they parallel the underlying principles that correspond to the three modes of socio-ecological resilience and, by consequence, to the uncertainty associated with climate change. Secondly, they achieve any one of urban resilience's three objectives as identified by Meerow et al. [16] across multiple temporal and spatial scales. And lastly, they constitute operational morphological determinants that, while universal, are applicable to a context regardless of its temporal or spatial scale. Among the sorted concepts, we identify six concepts that are most pertinent to urban resilience and meet this paper's objectives: ecological sensitivity, indeterminacy, polycentrism (de-centrality and modularity), connectivity (permeability), functionality (polyvalence), and redundancy. Fig 1 illustrates how these concepts situate within the wider knowledge of resilience, integrating climate change discourse and urban environments. Table 1 also summarizes their sample variables.

Beginning with *ecological sensitivity*, which emerged first in Ian McHarg's [50] *Design with Nature* and later in the theory of landscape ecological urbanism [51]. Similar to urban

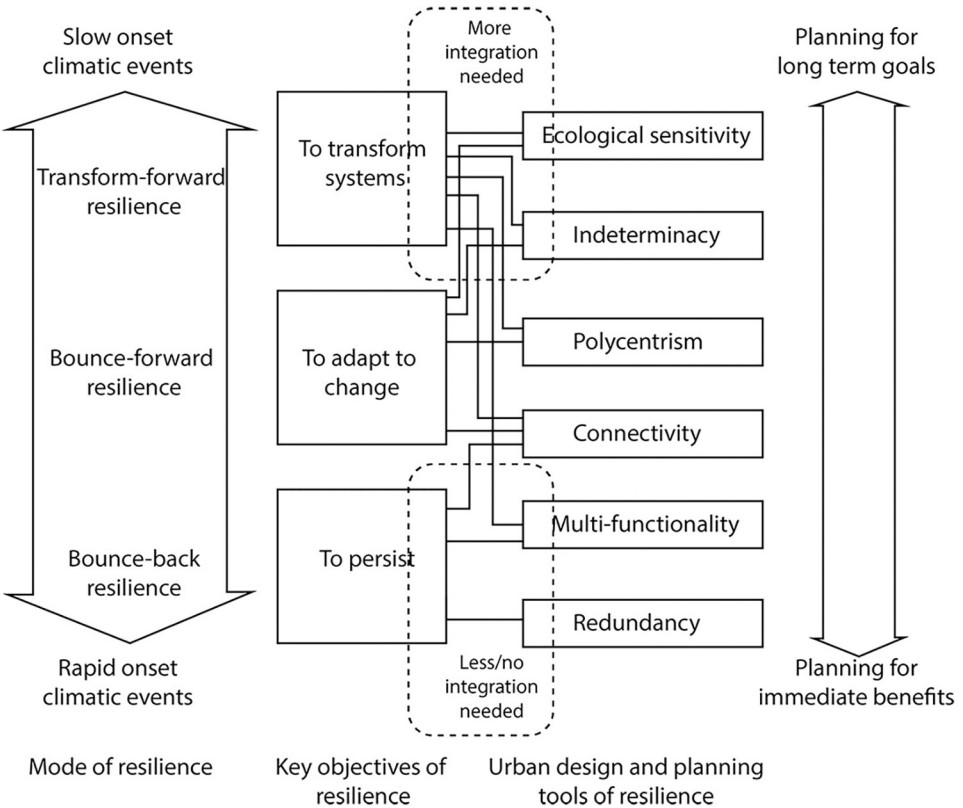

**Fig 1. The relation between the proposed urban morphological concepts and resilience.** This figure also illustrates the uses/effectiveness of these concepts both in emergencies and during slow-onset events. In other words, concepts relatively at lower levels of the framework may facilitate emergency responses with no/little support from others. But concepts at the upper levels (in tackling slow onset events and promoting transformation) need more integration of these concepts.

resilience and paralleling Beatley and Newman's [31] biophilic cities and Quigley et al.'s [29] socio-ecological resilience, ecological sensitivity prioritizes either enhancing and/or avoiding lessening a system's natural ability to survive disruptions when it responds to "temporal change, transformation, adaptation, and succession" [52, p.39]. Likewise, Smithson's [53] mat urbanism promotes a generative modification of urban form to gradually integrate natural and human systems while simultaneously considering the landscape itself as the infrastructure [54]. Moreover, mat urbanism underscores the connectivity among local ecosystems (e.g., networks of connected patches) and the flexibility in planning (e.g., clusters with overlapping programs and a range of functions that alter over time) (see also [55]). More recently, landscape ecological urbanism [56] operationalizes this concept at the urban scale by considering the landscape's ecological components as the constituents or the fundamental blocks of urban development as opposed to land uses and the town plan elements (streets, blocks, and building footprints). Such morphological compositions that integrate ecological sensitivity and ecological resilience, hence, yield "ecologically sound" urban developments that parallel "ecologically sustainable" developments [57, 58]. Examples of integrating ecological sensitivity include combining local eco-infrastructure and land reclamation to develop safe-to-build zones [15].

The second concept is ***indeterminacy,*** which refers to urban form's inherent ability to consider the future as unknowable, uncertain, and variable, hence, to cater for future needs and technical upgrading. Indeterminacy, which parallels Besim Hakim's [30] generative urbanism

**Table 1. The six concepts, indicators, and variables of resilient urban morphology.**

| Concepts | Brief descriptions | Sample indicators | Sample variables | Sources | |
|---|---|---|---|---|---|
| | | | | Socio-ecology | Planning and design |
| Ecological sensitivity | Integrates ecological systems within built form and ensures the health and vitality of ecological systems | • Conservation of species and ecosystems<br>• Increase of urban porous surfaces<br>• Increase in urban green and blue spaces | • The areas designated for conservation and/or ecological restoration<br>• The integration of blue and green infrastructure within built form<br>• The ratio of building footprints and impervious surfaces | [85, 86] | [15, 50, 69] |
| Indeterminacy (openness) | Underscores design flexibility that increases urban form's ability to transform over time in response to climatic uncertainty | • Urban form's ability to transform and change as needed<br>• Design potential to identify urban form's ability to serve unknown, but arising, needs. | • The type and range of potential functions available<br>• The number and/or area of spaces with unrealized potential (leaving room for future, yet unknown, needs)<br>• The number, shape, and spatial distribution of urban form elements that may be changed/adapted (physically or functionally) | | [60, 77, 87] |
| Polycentrism (distributed or modular system) | Discourages centralized systems and promotes distributed systems to spread out risks spatially and temporally | • Decentralized and distributed urban services and land uses,<br>• Design potential (e.g., streets) to connect/disconnect these services<br>• Degree of the polycentric system (e.g., waste, water, and energy management) | • The number of distributed services,<br>• The nature and degree of their distribution (spatial configuration)<br>• The number and spatial distribution of urban form<br>• The composition of urban form's modules | [68] | [67, 78, 88, 89] |
| Connectivity (accessibility) | Promotes permeability and facilitates emergency evacuations | • Multiple connectivities of street-networks and small street,<br>• Alternatives to during emergency, e.g., evacuation routes<br>• The access to urban amenities, evacuation paths, and cyclone centers | • The number of cul-de-sacs (the smaller, the better),<br>• The size (dimensions and area) of urban blocks (the smaller, the better)<br>• Travel distances, modes, and times to urban amenities, evacuation paths, and cyclone centers | [85, 86, 90] | [23, 69, 91] |
| Multi-functionality (polyvalence) | Highlights the potential to redesign urban form and allows for a multiplicity of uses when necessary | • Multiple-use over time in different situations<br>• Heterogeneity in urban form<br>• Mixed land uses | • The range of diversity and mixture in land and building uses (spatially and temporally)<br>• The number, range of diversity, and spatial distribution of existing and potential future uses of buildings and spaces<br>• The opportunities to accommodate temporary activities based on arising needs | [68] | [69, 92, 93] |
| Redundancy (diversity) | Encourages multiple systems or alternatives to similar functions | • Number of alternative adaptations available<br>• multiple street networks and evacuation paths<br>• redundancy of urban services and amenities | • The number, area, range of diversity, and spatial distribution of uses for open spaces (e.g., streets) and buildings especially, those needed in emergency situations,<br>• The number, area, range, and spatial distribution of spaces and buildings that offer alternative options for social and hard infrastructure services | [68, 94] | [23, 69, 79, 91, 95] |

and Vale's [17] progressive resilience, emerged after WWII in response to the rigidity of typical modern architectural layouts in mass housing with which end-users were dissatisfied [59]. Habraken's [60] "open architecture" also promoted indeterminacy and incorporated fixed and durable elements (i.e., supports) along with undetermined, less durable and changeable (i.e., in-fills) ones that may be adapted as needed over time (see also [61]). Indeterminacy is achieved through flexibility and adaptability as marks of a good fit between urban form and function [62]. For example, Stanford Anderson [63] operationalizes indeterminacy by

identifying a triad of urban street domains based on function: those with exploited potential (according to current needs), those with recognized but unexploited potential (already planned for the future), and those with unrealized potential (leaving room for future, yet unknown, needs). The latter two particularly pertain to uncertainty whereby their morphological integration enables urban form "to adapt to the irreversible, unpredictable, and ongoing changes" [64] (p.334). Examples include bioswales along streets, as part of an urban landscape, which can also manage increased stormwater runoff [65].

The third concept is *polycentrism* which promotes decentralized and distributed modular systems that disperse the risk. A polycentric system breaks its chain of reaction leading to damage containment rather than spreading to the whole system. Polycentrism is widely applied in landscape architecture, but when applied to urban design, it ensures a "safe-to-fail" approach that contributes to the resilience of urban form [66, 67]. Specifically, modularity generates independently-run functions within the various parts of a polycentric urban form in response to various climatic hazards [68, 69]. Indeed, empirical studies reveal increased efficiency in both movement (e.g., commuting and evacuation) and ecosystem services (e.g., thermal comfort), especially when polycentrism is combined with ecological sensitivity [70, 71]. Moreover, decentralized infrastructure systems that are based on modules and microgrids (e.g., energy, water, and sewerage facilities) are more climate resilient than centralized ones [72], albeit with the caveat that such systems might be costly.

The fourth concept is *connectivity*. Increased connectivity (through the street network) escalates permeability (the possibility of getting from place to place differently). Finely meshed gridded street networks maintain strong connectivity and, by consequence, permeability allowing easy and quick movement among smaller-sized urban blocks. For example, Azhdari et al.'s [73] experience in Shiraz, a semi-arid city in Iran, confirms that smaller parcels contribute more to lower surface temperature than the new zones with large blocks. These small blocks also enhance the integration among the various modes of movement (i.e., allowing transit, cycling, pedestrian, and automobile to co-exist more efficiently). Such diversity of movement choices in the case of evacuations during emergencies leads to decreasing risks (particularly to human assets). The hierarchies, curvilinearity, and cul-de-sacs of laddered street networks render them hard to navigate during emergency evacuations and significantly challenging to modify if needed. Empirical evidence from the historic quarters of Latin American cities [74] and from Glasgow [18] reveal that the connectivity of finely meshed grids leads to permeability, hence, significantly reduces evacuation times more so than laddered street patterns. They also lead to a distribution of smaller-sized urban blocks that increase the accessibility to resources (e.g., shelters, food, and health services), thus, enhancing climate resilience [18].

The fifth concept is *multi-functionality* (also known as *polyvalence*). It facilitates redesigning components of urban form (e.g., streets and open spaces) to accommodate diverse functions according to arising needs [75, 76]. Polyvalence aligns with Jabareen's [5] risk city, Hakim's [30] generative urbanism, and Vale's [17] progressive resilience. For example, Roggema et al. [77] identify polyvalent spaces throughout Dutch cities, marked as "unknown" spaces, that may accommodate various functions, like debris collection and emergency shelters, during and immediately after rapid onset climatic events. Similarly, Moudon's study [78] reveals that when San Francisco's irregular streets intersected, they yielded spaces with unrealized potential that also offer "breathing spaces" (i.e., recreational) to the surrounding communities. Indeed, an urban block that may alter its designated use depending on immediate or future needs (e.g., during a climatic hazard) contributes to improving climate resilience [69].

The last among the concepts is *redundancy*, which encourages several alternatives for the same activities to increase a system's opportunity to run during an emergency [23].

Redundancy parallels notions from Quigley et al.'s [29] socio-ecological resilience, whereby socio-ecological literature often considered redundancy as a backup plan or an unnecessary repetition in a system [68]. But, in urban design, redundancy entails functional and spatial diversity that leads to a certain level of excess through the presence of multiple systems (e.g., streets, plots, infrastructure supply nodes, etc.) that contribute to urban form's complexity [20]. Hence, it provides convenient alternatives to access services/functions while minimizing movement/travel time during emergencies [5, 79]. Redundancy, for example, occurs when there is a capacity/opportunity for land division into plots and parcels, thereby increasing urban form's diversity [80]. Such diversity facilitates, in a situation of high uncertainty, the creation of buffers (e.g., to stop the use of the most vulnerable plots) or the opening up of other alternatives (e.g., increasing the number of evacuation paths and plots). Evacuation studies after the 2010 and 2011 Christchurch earthquakes reveal that redundant alternative streets presented a more flexible and adaptable network through and around the city during the recovery phase [81]. Mixed land uses (horizontal) and mixed building floor uses (vertical grain) increase functional and spatial diversity. Specifically, places with easy-to-evacuate functions on the ground level decrease the risks to humans and assets in the case of a rapid-onset disaster (e.g., storm surge) in urban areas [65], (also see [82]).

This discussion highlighted the analogies between these six design concepts and the socio-ecological resilience. We acknowledge that beyond these six, other concepts may also contribute to enhancing and assessing urban resilience. Several authors, Allen [62], Jabareen [5], Quigley et al. [29], and Pickett et al. [83], for example, identified several similar concepts. Our review and discussion particularly highlight those that pertain to urban design and that are used to assess urban form's climate resilience. Our six concepts also align with the IPCC's [84] recommendations of enhanced adaptive capacity and urban resilience. They also balance the reactive/proactive and short/long-term adaptation responses. We connect these concepts to the three resilience modes and Meerow et al.'s [16] three resilience objectives (Fig 1). Then, we construct a theoretical framework in Fig 1, the concepts of which we operationalize in Table 1. In the latter, we identify the indicators and the variables for each concept to facilitate the assessment of urban form's resilience.

## 3. Materials and methods

### 3.1. Geographic focus

This study applies the six urban design concepts to Long Bay, also dubbed the Seven Miles Beach, in Negril, a resort town located on Jamaica's northwest coast. Negril is a small town with an approximate population of 8000 inhabitants, and falls between two Jamaican parishes: Hanover and Westmoreland (Fig 2). Negril's tourism industry contributes over 5% to Jamaica's national economy, whereby most of this industry and the associated tourism activities are concentrated in Long Bay. Like other coastal cities across the Caribbean and beyond, Negril's tourism industry is threatened by climate-change related hazards and the associated risks (particularly beach erosion). Within Negril, Long Bay is the area that is sandwiched between the Caribbean Sea and the Great Morass, hence eliminating the possibility of retreat. Long Bay's vulnerability is exacerbated by its ribbon-like settlement pattern characterized by a high density of resorts and small building footprints [96]. Consequently, Long Bay is at high risk from a combination of climate-change related impacts, particularly late season tropical storms and hurricanes (e.g., hurricanes Mitch, Michelle, and Wilma) [97] as well as beach erosion.

A recent report by the World Meteorological Organization (WMO) [98] estimates that between 1993–2020, the sea level in the Caribbean has been rising at a rate of 3.6 mm per year, slightly higher than the global average. Another study by the Maritime Provinces Spatial

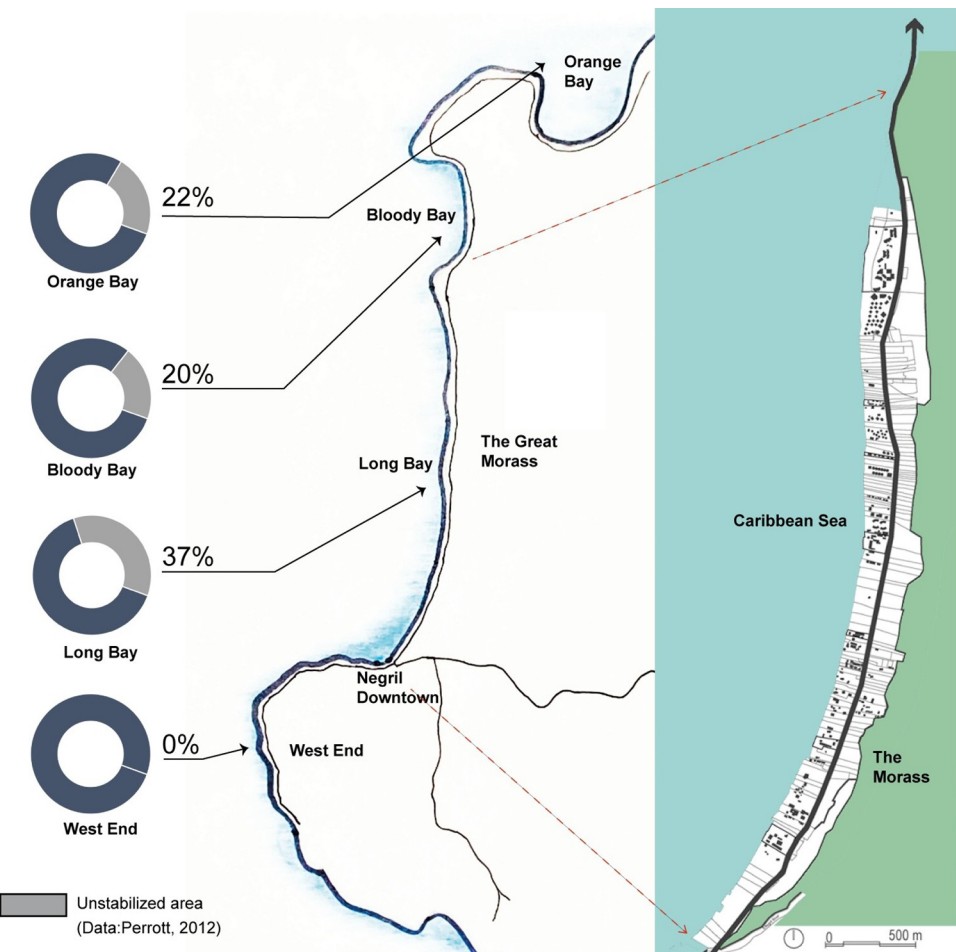

**Fig 2. The location, geological profile and ribbon-like urban morphology of Negril and Long Bay (Authors draw the map based on a few references from Google map).**

Analysis Research Center and Saint Mary's University [99] compares the geological profile of the Negril area and found that about 37% of Long Bay's land is unstable or partially stable higher than the surrounding area. Perhaps that is why between 43% and 91% of Long Bay's beach erosion is only caused by sea-level rise leading to a loss of up to 1.4 metres/year–a higher rate than its neighbouring areas like Bloody Bay and Orange Bay [100] (Fig 3). Several governmental and non-governmental organizations (NGOs) at different institutional levels maintain diverse and often differing opinions on the most suitable adaptation strategies for Negril. Yet, these strategies share the commonality of mostly focusing on coastal protection. In this study, we investigate these strategies—whether they have been implemented or proposed for the future—and assess their influence on the resilience of Long Bay's built form. We chose Negril, as a single case, to test our framework for two reasons based on Yin's [101] criteria. The first is *significance*. Negril symbolizes the critical case and experiences the real challenges of climate change, similar to other small islands and coastal settlements worldwide. This study helps us verify the applicability of the framework. The second is Negril's *uniqueness* in terms of its geographic and built environment features. Specifically, Negril's unique ribbon-like urban morphology remains still unexplored from urban design investigations.

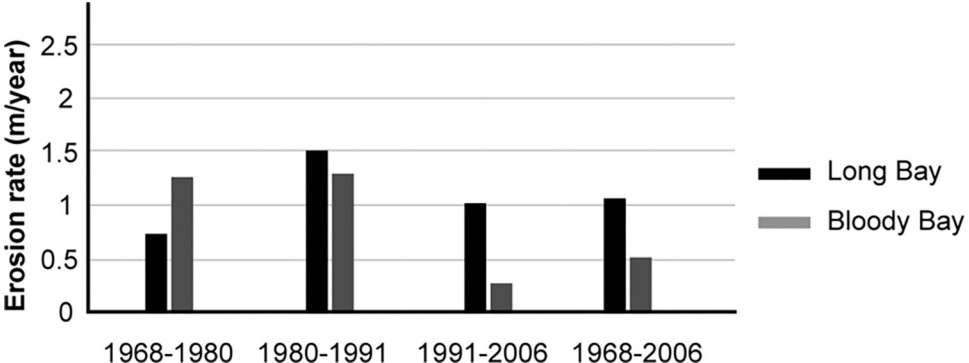

**Fig 3. A comparison of the rate of beach erosion between Long Bay and nearby Bloody Bay (adapted from [100]).**

## 3.2. Data sources

This study used empirical and qualitative evidence that was carried out in two phases: in situ documentation of the urban morphology and the ecosystem conditions in Long Bay and conducting a total of 19 semi-structured in-depth interviews with key informants. We obtained the verbal consent of our informants before interviewing, and all data were analyzed anonymously. The Office of Research Ethics at the University of Waterloo, ON, Canada, approved this study. The approval number was ORE 20830.

**3.2.1. In situ documentation of the urban morphology and ecosystem conditions.** The first phase entailed the documentation of Long Bay's urban morphology and the conditions of its ecosystem through in situ observations. This phase was carried out for 14 days in May and June 2014 and involved a team of 10 researchers working throughout this period. A follow-up observation in August 2015 to map and validate specific information obtained from interviews (discussed in section 3.2.2.). The team conducted transect walks [102] to observe and document Long Bay's morphological conditions through mapping, notetaking, and photography. The team also obtained secondary data like planning and policy documents from an array of organizations (such as the National Land Agency, Jamaica). Urban morphological data (consisting of the existing street network, parcels, and building footprints) were obtained from the Mona GeoInformatics Institute at the University of the West Indies, Mona Campus. Additionally, casual interactions with local inhabitants and tourists during the fieldwork added a wealth of information to contextualize and validate the primary observational data of the challenges facing Negril's built environment and its existing adaptation strategies.

**3.2.2. Key informants' insights on Long Bay's adaptation planning.** The second phase, which spanned over two weeks in August 2015, entailed conducting a total of 19 semi-structured in-depth interviews with key informants who hailed from two levels of governance: macro and micro. Jamaica's governance system is centralised as a small island developing state (SIDS). The head offices for all the government agencies, which are based in the capital, Kingston, make all the planning and development decisions for the entire island. Thus, at the macro level, this study commenced with interviewing key informants, i.e., professionals who work for these central government agencies. This was followed by interviews at the micro, local level of governance in Savanna-la-Mar and Negril. The key informants included: planners, academics, environmental and tourism experts, and other professionals who are involved in climate change-related planning and development for Negril. With their experiences and in-depth knowledge, they facilitated achieving this research objective to understand and assess Negril's resilience capacity. The recruitment process of interviewees started during the first

**Table 2. The agencies interviewed for this study.**

| Agencies | Agency type | Head office (Kingston) | Local office |
|---|---|---|---|
| Jamaica Hotel and Tourist Association | Non-Government | | ✓ (Negril Chapter) |
| Ministry of Water, Land, Environment & Climate Change (MWLECC) | Government | ✓ | ✓ (NIGALPA) |
| National Best Community Foundation | Non-Government | ✓ | |
| National Environment and Planning Agency (NEPA) | Government | ✓ | ✓ (Enforcement branch, Negril) |
| Negril Area Environment Protection Trust (NEPT) | Non-Government | | ✓ |
| Office of Disaster Preparedness and Emergency Management (ODPEM) | Government | | ✓ (Westmorel and parish) |
| Urban Development Corporation (UDC) | Government | ✓ | |

phase and continued through June and July of 2015. We identified the relevant agencies and their divisions first through reviewing institutions and their websites (e.g., the Planning Institute of Jamaica) and then through consulting with the researchers' academic and professional networks in Jamaica and the Caribbean, including planning and environmental professionals, NGOs, academics, journalists, and environmental activists.

Initially, about 25 key informants were identified and invited by phone and/or email to participate in the study. During the second phase in August 2015, only 16 participants were interviewed based on their availability and asked to recommend experts who could also be interviewed for this research project. Accordingly, three additional key informants were identified and interviewed. It is worth noting that all key informants were selected based on their designation and affiliations; we tried to reach the most experienced personnel (based on seniority with over-10-year working experience in the field). In case of their unavailability, we searched/invited the next available senior person to avoid the effect of "chain referral", a common limitation of the snowball sampling [103]. Thus, in total, we held 19 interviews with professionals with an average professional experience of 16 years. Table 2 lists the respondents' primary affiliations only since some are affiliated with more than one institution/agency (e.g., the Planning Institute of Jamaica and NEPA).

Each in-person interview lasted for nearly an hour and ceased when the information got saturated. Primarily, each interview included discussions about: i) the climate change risks and their causes in Negril (the combination of hazard, vulnerability, and exposure), ii) the current and future initiatives of adaptation planning and the challenges facing such initiatives, and iii) how each of the six urban design concepts could potentially be applied to assess the resilience of Long Bay's built form.

## 3.3. Data management and analysis

For all data collected for this study, we set three a priori themes (that became the codes) for data management and interpretation: i) the climatic risks to Long Bay's built environment, ii) the existing adaptation plans for Long Bay's urban form, and iii) the potential of applying the six urban design measures to Long Bay's urban form. These themes constituted the main headings under which subthemes/subheadings emerged [104]. The themes/subthemes represent "a pattern in the information that at minimum describes and organises the possible observations and at maximum interprets aspects of the phenomenon" [105] (p.161). They also led to generating subthemes, such as beach erosion and the degradation of ecosystems under the risks theme and so on.

The data collected during the first phase (in situ documentation of the urban morphology and ecosystem conditions through mapping, notetaking, and photography) were digitalized using relevant software packages (e.g., AutoCAD and Adobe Illustrator). The results were then

plotted into a matrix with a continuum that relayed, for instance, the rate of increase or decrease in vulnerability and its causes. The interviews conducted during the second phase were each audio recorded and transcribed. The contents of transcripts were then analyzed and grouped under the three themes. Such data classification process facilitated managing the qualitative interview data (over 19 hours of recording) into manageable themes and sub-themes around the three discussion topics [106].

We triangulated the data from: 1) the in situ documentation of the urban morphology and ecosystem conditions (including our casual conversations with local inhabitants); 2) the in-depth interviews with key informants; and 3) the urban morphological data (the GIS data). This enabled us to crosscheck the findings to ensure the validity of the data.

## 4. Results and discussion

### 4.1. Ecological sensitivity

Referring to the Town and Country Planning Provisional Order for Negril [107], a planner in a public agency argued, "we consider Negril to be an ecologically sensitive area, and the design would/should be consistent with the sustainable development. . . and biodiversity of the entire area." However, due to the degradation of the marine ecosystems in Negril generally and Long Bay particularly (e.g., mangrove forests and coral reefs), their natural ability to attenuate wave energy and replenish sand beach remains low. Reinforcement of upper watersheds to enhance ecological connections among rivers and channels to the sea would help coastal flood management [15]. An environmental expert also recommended "eco-sensitive approaches. . . [that can] restore and/or rehabilitate coastal ecosystems, sea grass, coral reefs, and other coastal vegetation that would build resilience" and, thus, enhance (natural) adaptive capacity. A few resorts have already adopted ecosystem-based adaptations that integrate ecosystems and their services within the area. Our findings from our observations confirmed that Sandal Resort provided substrates for artificial reefs. Most respondents agreed that the existing small building footprints generate a higher ratio of open and green spaces conducive to decreasing surface rainwater runoff and incorporating local plants such as sea grapes (*Coccoloba uvifera*) and blue mahoes (*Hibiscus elatus*), and local mangroves that were observed widely along Long Bay. Indeed, local inhabitants were aware of these plants' benefits as they cut wind energy during storms. During the casual conversation with the locals, they showed us how they tried to grow mangroves in plastic pipes to increase natural capacity to reduce erosion. All interviewees also highlighted the strengths of local ecosystems and their incorporation into landscape design and planning. For example, integrating street bioswales will increase rainwater filtration, minimize rainwater surface runoff, and alleviate flooding impacts and pollutants flow during heavy rains.

### 4.2. Indeterminacy

Indeterminacy refers to the flexibility of urban form, particularly its innate potential to reorganize and transform so as to cope with uncertainty. A respondent shared her experience: "well, I remember, after the tropical storm Nicole [in 2010], there was a significant amount of beach debris. . .washed off in all beaches across Negril. . .properties didn't have experience, and they didn't know what to do." We also observed that the public community hall located nearby Negril Planning Authority's offices co-functions as an emergency shelter for the entire area, while Long Bay's two public beaches (located at the southern and northern ends) could potentially accommodate temporary shelters for people at risk and support recovery actions, including piling up of seaweed and debris (Fig 2). Still, one local planner was dissatisfied with these spaces' indeterminacy potential, arguing: "I don't agree that there is enough room to facilitate

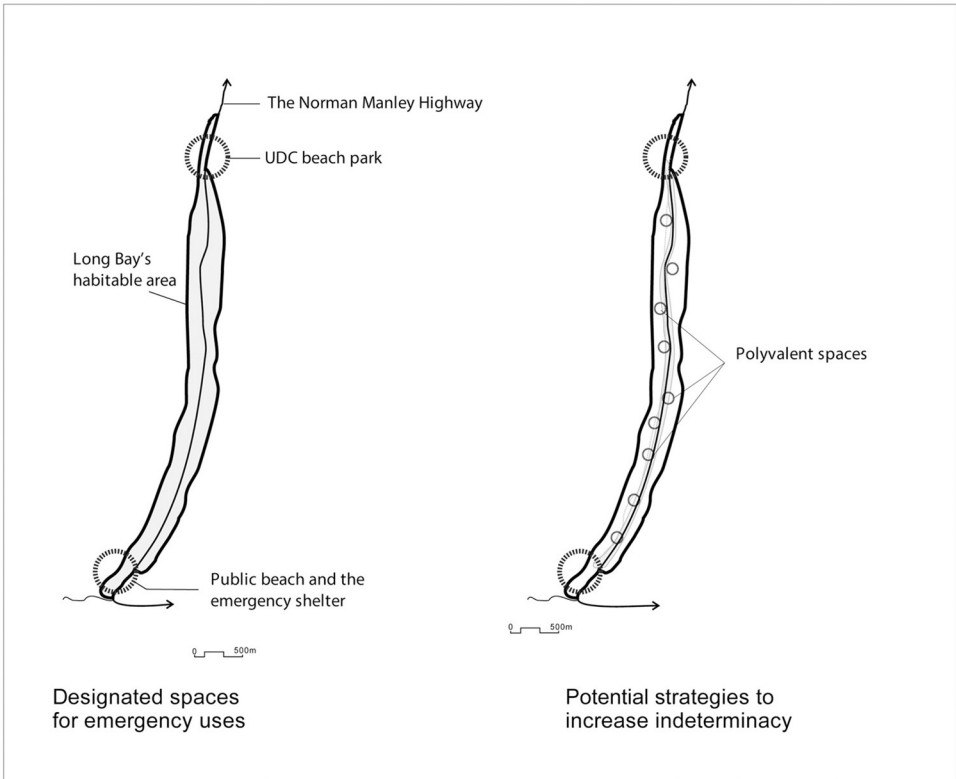

**Fig 4. Current indeterminacy of Long Bay's and its potential.**

such emergency responses." Yet, a few other key informants highlighted that "for Long Bay, it is important when expecting any adverse impacts . . . [to] call for evacuation to survive." Thus, our interviewees often recommended that every hotel should prepare plans and policies for emergency management, like shelters and evacuation paths, which should be integrated into the government's plans for future disaster risk management.

Indeed, we observed that a few new hotels, like the Travelers Beach Resort and Sandals Resort, applied indeterminacy in their common spaces and lobbies that are above 3.2m from the average sea-level which is set as a projected storm-surge level for Long Bay. By doing so, these hotels allowed their common spaces to be used during an emergency. There are also other spaces throughout Long Bay that fulfill indeterminacy in that they may be converted into a variety of uses as the need arises during an emergency. These places include, among many others, two public beaches at the northern and the southern tips of Long Bay (see Fig 4), privately owned open spaces along Norman Manley Boulevard, and the current coastal setback along Long Bay.

### 4.3. Polycentrism (distributed systems)

Polycentrism refers to distributed systems that spread risk through the system. A distributed system enables water and energy infrastructure, for example, to run independently and separately within a spatial zone or module. Our findings from observation and maps reveal that Negril's urban infrastructure and services (e.g., electricity, waste, and water) are centrally managed. Their vulnerability is exacerbated because of Long Bay's ribbon-like morphology, along which all these services and infrastructures run parallel to Norman Manley Boulevard.

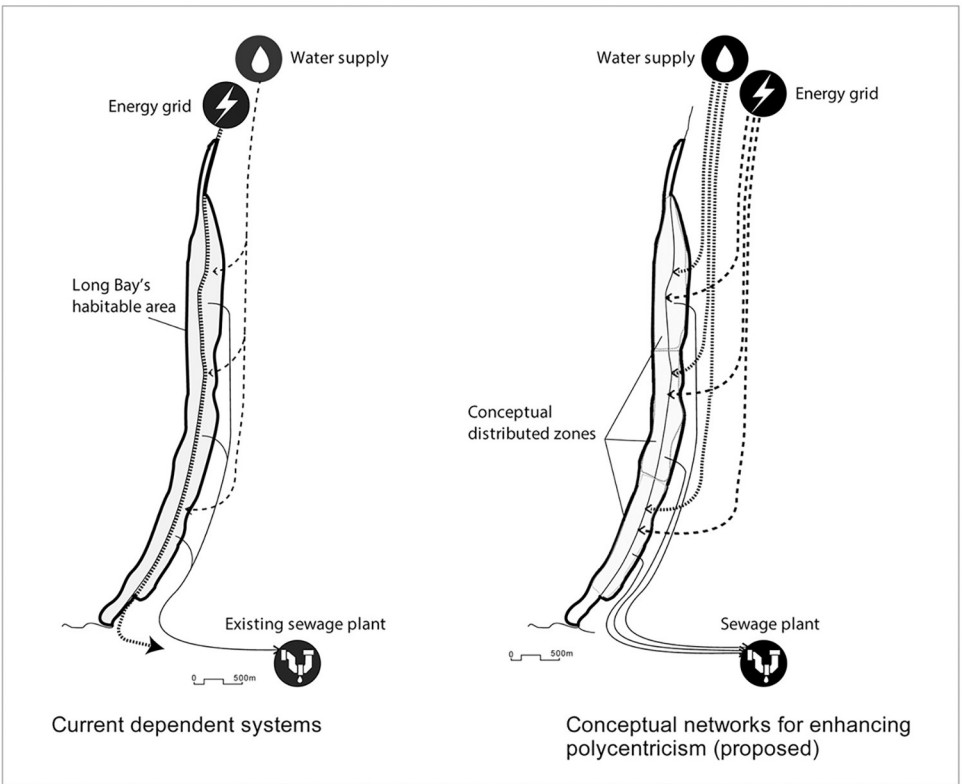

**Fig 5. Current polycentrism of Long Bay and its future potentials.**

Therefore, even partial damage to or loss of any of these services and infrastructure may result in the collapse of the entire system in Negril. For example, in 2010, a massive flood blocked a part of Norman Manley Boulevard and left all of Negril and its surrounding region entirely inaccessible for ten days. During interviews, a few key respondents argued that with additional infrastructure and service networks, Norman Manley Boulevard might be divided spatially into smaller zones/modules that can function independently from each other during emergencies. One of our respondents argued, "a distributed system can reduce the risk of climate change. . . [if] you have a centralized sewage. . .or source of drinking water, for example, and you have a major hurricane or something that destroyed, I and you won't have any alternative" (Fig 5). However, most respondents from public agencies recommended maintaining a centrally managed sewage system (as in the nearby town of Sheffield) to reduce operational and maintenance costs. One respondent from a governmental agency argued that "we built a centralized sewage system. We are forwarding toward centralized not much toward distributed systems. . .[because] we have limited useable land."

## 4.4. Connectivity

Connectivity (through the street network) leads to increased permeability (the possibility of getting from place to place in different ways). Long Bay's ribbon-like form, sandwiched between the Caribbean Sea and the Great Morass, yields, spatially, a linear axis that runs north-south along a single street (Norman Manley Boulevard), leaving no room to accommodate any other north-south streets. During interview, a disaster risk management expert insisted that "roads are visually and physically accessible for the most part..[but] access to the

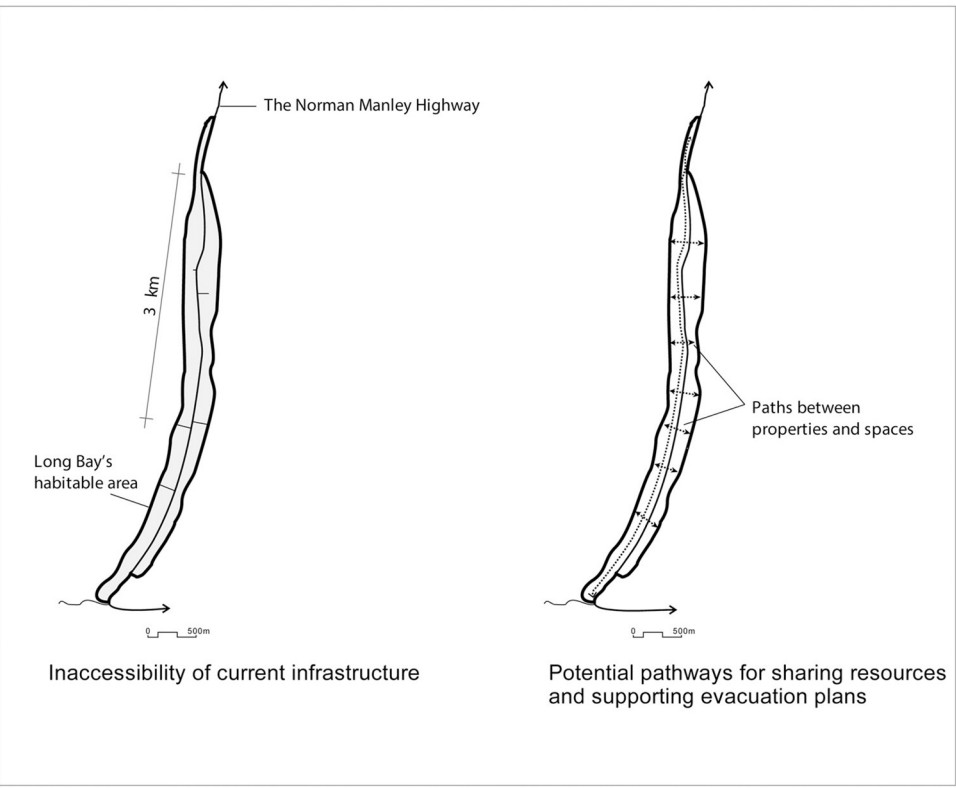

**Fig 6. Connectivity of Long Bay's street networks.**

beach is fairly poor. . .[property] boundary walls likely contribute to incidences of localised flooding." Accordingly, the respondent recommended alternative movement options for pedestrians and cyclists to increase permeability, namely public paths and walkways parallel to Norman Manley Boulevard. Also, a few existing transverse alleyways between the buildings would increase the permeability between the boulevard and the beach. One key informant suggested that "private [property/hotel] owners run small business [e.g., bars, craft shops] and should also welcome visitors/tourists [from the boulevard] to use their properties." His suggestion also hinted at how the economic opportunities at the plot level (each private property) would lead to developing transverse connections from Norman Manley Boulevard to the beach. Thus, Negril planning can increase connectivity and, consequently, recovery actions and emergency evacuation (Fig 6).

## 4.5. Multi-functionality / polyvalence

Multifunctionality refers to incorporating, spatially and temporally, mixed land uses (horizontal) and building uses (vertical grain), and accounting for potential future uses in disaster-prone areas. Currently, the regulations support mixed land uses along Long Bay, including tourism, retail, and residential developments, even though small hotels dominate the area. Regarding the sea-level rise and its future impacts, most respondents argued that full retreat (i.e., the possibility of moving buildings/facilities inland away from hazard-prone areas) might be the crucial option when no other options are possible. The northern part of Long Bay represents what Robinson et al. [96] particularly identify as a 'hot spot' where beach erosion is expected to result in a loss of up to 25 metres of the beach by 2050. Another study [108]

estimated that only 1metre of sea-level rise would fully or partially damage 29% of coastal resorts in the Caribbean basin. As much as 55% of them are under threat of beach erosion. Weighing future risks against the vibrant beach tourism, our key informants agreed on the weak resilience of this hot spot. They suggested more temporary modes of functions than permanent ones: "I am looking for a reduction of residential uses [but more] commercial, probably for resort or for that type of mixed. . . But in the long run though, [Long Bay] might be more recreational/leisure [and] you will have less occupancy. . . [but only have] a beach, more bars, and commercial. So, tourists would probably be elsewhere on the hills and come down during the day to whatever activities are there [in Long Bay]." This means that tourists would enjoy the recreational activities along the beaches only during the daytime, in order to address the safety concerns associated with rapid onset climatic events, hence, avoid the loss of lives. Accordingly, the current land use patterns and future planning and development policies should be adaptive and flexible to cope with future needs and uncertainty while simultaneously considering the area's carrying capacity.

## 4.6. Redundancy

Redundancy entails that urban amenities and land uses be designed to allow for backup plans and alternatives to essential services (e.g., shelters, clinics, and basic infrastructure like water and energy). Redundancy is particularly useful when dealing with the immediate impacts of rapid-onset events. Long Bay's ribbon-like morphology results in its dependency on Norman Manley Boulevard as the main movement artery. If even a small part of this boulevard becomes blocked or damaged during/after a disaster, it will impede disaster recovery and emergency evacuations. Nevertheless, the key informants instead focused on praising the redundancy of Negril's telecommunication systems, where landlines and mobile networks may substitute each other. They also highlighted how the government of Jamaica promotes the development of localized solar and renewable energy sources as alternatives to the national grid. Our observations and conversations with local community members also revealed that hotels like Sandals in Long Bay, Fiesta Hotel (in Hanover), and Rick's Cafe and the Rockhouse Hotel (in the West End) have also installed solar energy; the latter even maintains its own recycling system. Yet, fossil-fuel-driven generators continue to prevail in Long Bay and serve as backups for emergencies and everyday use. Although such generators increase redundancy, they emit greenhouse gases and thus represent maladaptation [109]. The key informants also concurred on the potential benefits of redundant systems to enhance resilience, such as wind energy and harvesting rainwater at community and household levels. Still, simultaneously, they warned of the costs these systems require. Fig 7 shows the single system (e.g., electricity and waste management) that Long Bay currently depends on and the future potential to improve redundancy.

The key informants also agreed with the need for redundancy, though some remained aware of Long Bay's morphological limitations due to its ribbon-like form. "In terms of income activity, it [Long Bay] has only one-[beach] tourism; for transportation, it has only one road. In case of sea-level rise, redundancy may be necessary. . . but you have to balance cost and benefits," a development expert argued. Also, as quoted above, connectivity and redundancy complement each other. Property owners would "run small business" like bars, restaurants, and craft shops which would allow transverse access from Norman Manley Boulevard (i.e., enhance connectivity) while simultaneously providing redundancy in functions (e.g., food and shelter services). Both of which are crucial for emergencies. Surely, the existing low-rise buildings along Long Bay, with their small building footprints, open spaces, and temporary beachfront structures, provide ample opportunities for mixed uses both horizontally and vertically,

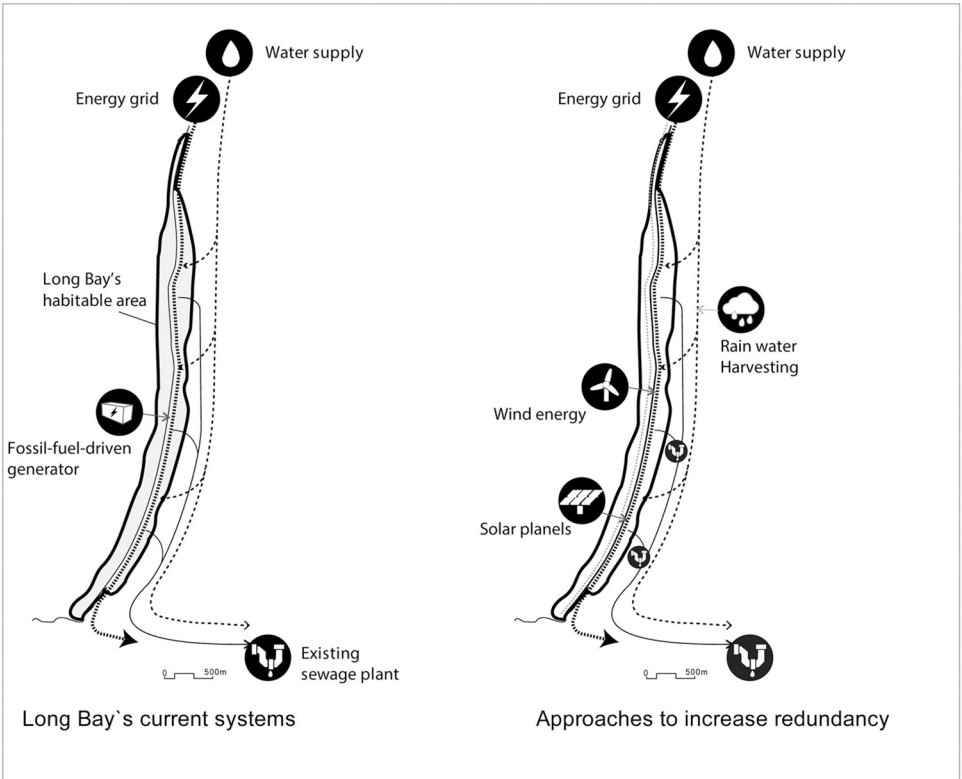

**Fig 7. The current infrastructure (e.g., energy and water) of Long Bay and ways to enhance redundancy.**

hence, contribute to redundancy. As discussed earlier, several hotels, like the Travelers Beach Resort and Sandals Resort, designed their lobbies and other communal spaces (that lie above the storm-surge level) as polyvalent spaces that can be converted from their current uses into emergency shelters when needed.

## 4.7. Negril's resilience and a path toward generative transformation

Our findings reveal that Negril's exposure is high due to its natural location (sandwiched between the Caribbean Sea and the Great Morass). It is also susceptible to a combination of rapid and slow-onset climatic events. Besides, its built form (ribbon-like morphology along a single spine, i.e., Norman Manley Boulevard) and the concentration of tourism-related activities collectively render it at high climate-related risks. Fig 8 summarizes the assessment of Long Bay's current resilience based on the six proposed concepts. It remains clear that Long Bay's unique morphological conditions cause the *redundancy* score to be low; however, local planners are aware of how it can be enhanced by providing alternative infrastructural or functional solutions. This figure also facilitates an understanding of the potential planning and design actions that need to be prioritised in any future action, which leads to the discussion of Long Bay's current planning and development proposals. While some of these concepts are new, some are emerging; a few are well-established but remain isolated. For example, *ecological sensitivity* has existed over decades in the design discourse [e.g., 50, 52, 110] and recently emerged through ecosystem-based or nature-based solutions to contribute to climate adaptations [111]. A comprehensive understanding of the framework is imperative to assess the resilience of urban forms. Surely, the framework reveals that rigidity, inflexibility, and/or centrality

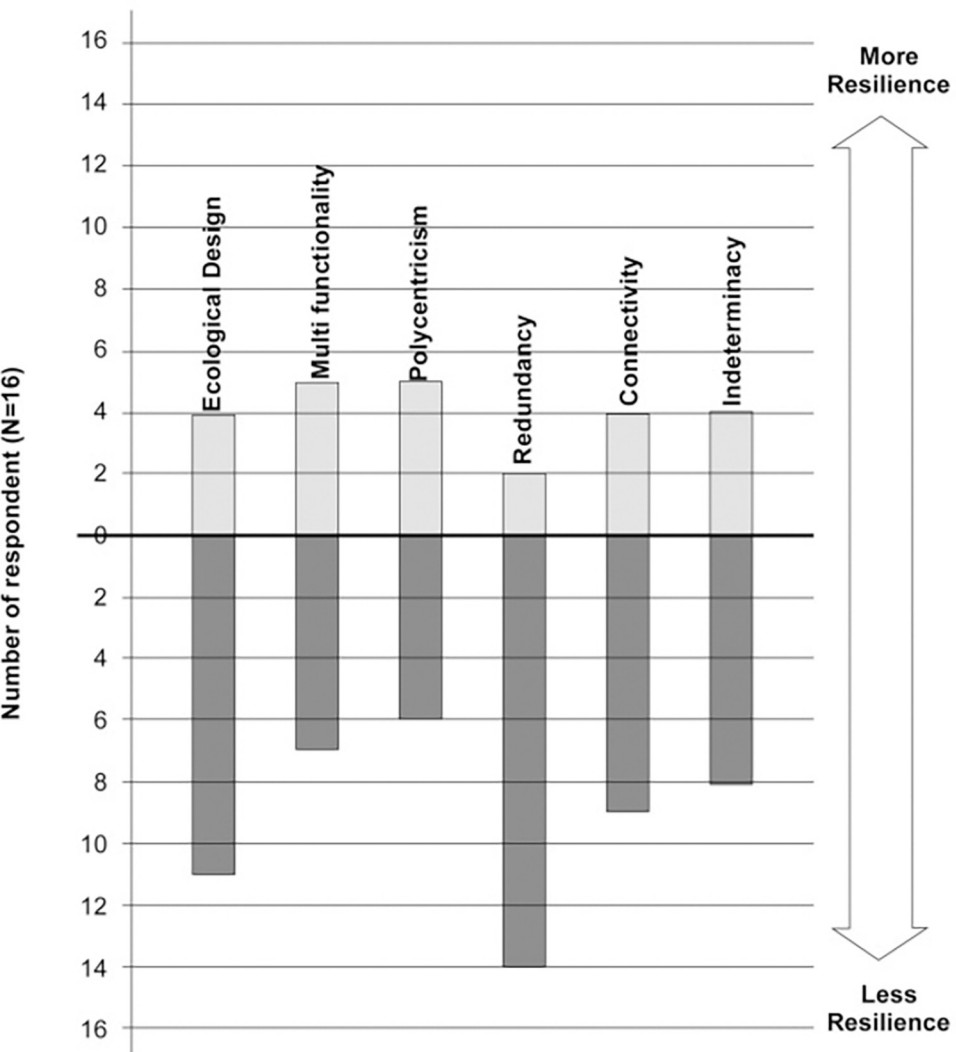

**Fig 8. The assessment of the current resilience of Long Bay.**

in urban design impede enhancing the resilience of a city/settlement to climate change. Our findings are also aligned with Roggema's [112], Sharifi's [24], and Feliciotti et al.'s [25] propositions. Also, they advance urban design applications to understand and assess the resilience of human settlements.

Global sea levels would rise up to 80 cm by 2100 and expect to displace 200 million by 2050 [113]. Coastal areas are particularly vulnerable to both rapid and slow onset impacts of climate change. However, many coastal adaptation plans, like Negril's, underscore strategies addressing only the rapid onset events (primarily from hurricanes and storm surges) and facilitate recovery actions (ex-post) [26]. In general, most risk reduction strategies in Latin America and the Caribbean still pay more attention to resisting and coping with severe disaster events, as Muñoz-Erickson et al.'s [114] experienced. Such bounce-back approaches and conventional design of urban form may protect Negril and other coastal settlements and their tourism industries, but only for a limited period. Many governments in the small islands remain reluctant to implement and/or enforce long-term, flexible, and incremental planning that would transform their current urban form to tackle future uncertainty–climatic and non-climatic [4,

114]. In contrast, Negril has approved more compact and high-density developments across Long Bay (by allowing additional floors to existing buildings) to increase tourism revenue. Many SIDSs, such as Bahamas [115] and Malta [116], experience similar challenges [113]. It is worth noting that climate adaptation initiatives sought to achieve a trade-off between economic opportunities and alleviating the immediate and future climate risks. In doing so, two things need to be assessed, as argued by our respondents and many scholars too: i) the inevitable future climatic scenarios and their local impacts informed by local informants and different scientific studies [e.g., 96, 117] and ii) the planning and design potential of urban form for long-term and incremental adaptations [118, 119]. The framework certainly eases such assessment. As our study reveals, Negril's current adaptation initiatives have hardly considered any of the two and, consequently, render more people and assets vulnerable to the impacts of climate change.

In a situation when many developing nations still lack accurate projections of climate change impact at the city/community levels, a long-term incremental planning approach is always recommended that can ensure the long-term benefits of economic opportunity and promote "low density and low impact development," as a respondent of a Jamaican public agency recommended. Theoretically, we consider such recommendations and dub them "generative transformation" because they resonate with Hakim's [30] generative development processes and with Vale's [17] progressive resilience in tackling future climatic uncertainty through urban design. Specifically, generative transformation refers to the gradual, progressive, and incremental changes to urban form, according to arising needs, to enhance its resilience based on various combinations of the six urban design tools. Most importantly, our study reveals a dire lack of systemic (and extensive) methods to assess the current resilience before making planning decisions regarding urban growth and adaptation plans–something that this paper sought to overcome by operationalizing the six urban design concepts.

## 5. Conclusions

Not only Jamaica, but most small island developing states (SIDS) across the world are highly vulnerable to climate change impacts. This paper presents an effort to understand the climatic risk in the context of Long Bay in Negril, Jamaica through the lens of six urban design concepts: ecological sensitivity, indeterminacy, polycentrism, connectivity, multi-functionality, and redundancy. and urban design potentials to alleviate the risks. We assessed 1) the current risks (combination of hazard, exposure, and vulnerability); 2) the existing and proposed formal adaptation plans; and 3) the potential for the long-term transformation of urban form to cope with climatic uncertainty.

Our interviews captured an in-depth representation of almost all agencies involved in climate change and planning for Negril in general and Long Bay in particular. However, we acknowledge the limitations of reaching a few agencies in Jamaica for this study (e.g., the Jamaica Environmental Trust). In addition to these agencies, politicians influence Negril's current and future adaptation planning decisions, but their voices remain outside of the paper's scope. This study assesses, as opposed to quantifiably measures, the climate resilience of the urban form of Long Bay in Negril. Quantifying each of our concepts requires both simulated morphological models and accurate projections of climate change-related data for Long Bay. Such data are unavailable for Long Bay and many other contexts in the developing world. Most developing countries at high climatic risks lack detailed and accurate data at the local scale yet require reliable methods to assess climate resilience and devise action. Although perceived as a quantitative shortcoming, the framework provides an alternative solution for such developing world contexts that struggle with the need to enhance climate resilience. Therefore,

our model provides planners and local activists, in the absence of accurate projections, with an alternative method to do so.

We acknowledge that the framework is universal, and its six concepts, along with their variables, must be incorporated in tandem (without excluding any). The interplay among the six will vary depending on each context's specific climatic risks (i.e., hazards, exposure, and vulnerability) and local particularisms (i.e., place identity). Thus, the framework needs professional interpretations and sensitive applications at the local level. For example, ensuring redundancy in certain types of infrastructure (specifically, waste and water) may be challenging due to Long Bay's ribbon-like morphology. Still, it is possible for others (e.g., generating electricity through solar panels). This shortcoming in Long Bay can be offset by increasing one or a combination of the other concepts to increase climate resilience, such as enhanced transverse connectivity and permeability, which also leads to increased tourist movement, hence, contributes to Long Bay's primary local economic activity. Though some aspects of Long Bay's morphology and local conditions are exceptional, it certainly is not unique. On the contrary, in many aspects, it is similar to most coastal areas that depend on a bounce-back resilience model that overlooks the ecological and the transform-forward modes of resilience (as shown in Table 1).

Whether in Long Bay or elsewhere in cities and towns, urban form is not easy to transform instantaneously due to a combination of cost and logistics. Therefore, our proposed "generative transformation" adopts a gradual, progressive, and incremental approach that can make adapting urban form to climate change more feasible and reasonable. Yet, to inform generative transformation, it is essential to assess the urban form's current resilience, hence, the significance of our proposed framework theoretically, empirically, and practice-wise.

First, *theoretically*, the underlying foundation of the proposed framework connects concepts discussed in the literature on urban design, socio-ecological resilience, and climate change adaptation. The framework provides a more transformative, anticipatory, and equitable approach to urban resilience to address the theoretical shortcomings in the interaction between socio-ecological sustainability and disaster risk reduction. But more importantly, the framework also introduces the notion of generative transformation to enhance the resilience of urban form while complementing other works in the field.

Second, *empirically*, these concepts provide a universally applicable urban design framework for resilience assessment that can be contextualized, as we have explored in Long Bay, to the specific conditions of a local context. Adaptation planning policies and actions in many developing states still heavily focus on specific risks and hardly incorporate long-term climate risks and uncertainty. The framework thus leads to incremental urban transformation to tackle long-term climatic uncertainty, particularly for those nations with scarce resources and limited climate change data.

And last, *practice-wise*, our proposed generative transformation with its sixfold framework provides an important, easy-to-use resource for professionals and community actors that facilitates their ability to assess the resilience of their community's urban form. Consequently, it recommends generating context-specific urban design guidelines and generative transformation solutions that acknowledge a community's unique potential and distinctive urban character.

## Supporting information

**S1 File.**
(ZIP)

**S1 Data.**
(XLSX)

**S1 Appendix. Indicators and variables to assess a community's resilience: A review.** (DOCX)

## Acknowledgments

We acknowledge the logistics and support provided by the Partnership for Canada-Caribbean Climate Change Adaptation (ParCA) and several organizations in Jamaica, including NEPA, NEPT, and CaribSave. We especially thank Ms. Jacqueline daCosta, the former president of the Commonwealth Association of Planners and the Jamaica Institute of Planners. Our deepest gratitude goes to all individuals across Jamaica who participated in the interviews and other forms of the field survey. Lastly, our deepest gratitude to the reviewers and editors for their constructive feedback.

## Author Contributions

**Conceptualization:** Tapan Kumar Dhar, Luna Khirfan.

**Data curation:** Tapan Kumar Dhar.

**Formal analysis:** Tapan Kumar Dhar.

**Funding acquisition:** Luna Khirfan.

**Investigation:** Tapan Kumar Dhar.

**Methodology:** Tapan Kumar Dhar, Luna Khirfan.

**Project administration:** Luna Khirfan.

**Resources:** Luna Khirfan.

**Software:** Tapan Kumar Dhar.

**Supervision:** Luna Khirfan.

**Validation:** Tapan Kumar Dhar, Luna Khirfan.

**Writing – original draft:** Tapan Kumar Dhar, Luna Khirfan.

**Writing – review & editing:** Tapan Kumar Dhar, Luna Khirfan.

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
