## [Decision Letter · Decision Letter 0]

1 Aug 2022

PONE-D-21-17612A sixfold urban design framework to assess climate resilience: generative transformation in Negril, JamaicaPLOS ONE

Dear Dr. Dhar,

Thank you for submitting your manuscript to PLOS ONE. After careful consideration, we feel that it has merit but does not fully meet PLOS ONE’s publication criteria as it currently stands. Therefore, we invite you to submit a revised version of the manuscript that addresses the points raised during the review process.

We look forward to receiving your revised manuscript.

Kind regards,

Dr. Md Nazirul Islam Sarker

Academic Editor

PLOS ONE

Journal Requirements:

2. During our internal checks, the in-house editorial staff noted that you conducted research or obtained samples in another country. Please check the relevant national regulations and laws applying to foreign researchers and state whether you obtained the required permits and approvals. Please address this in your ethics statement in both the manuscript and submission information."

VCR(EO)28May 21: Please at PRTC send back to Author (for confirmation of approval)  as follows:

"Thank you for including your ethics statement:  "Office of Research Ethics, The University of Waterloo, Waterloo, ON, Canada.

Approval: ORE # 20830

The data were analyzed anonymously".   

Please amend your current ethics statement to confirm that your named institutional review board or ethics committee specifically approved this study. 

3. Thank you for stating the following financial disclosure: "This work was supported by the Partnership for Canada-Caribbean Climate Change Adaptation (ParCA) (http://parca.uwaterloo.ca/). The grant number is:  106372-006"

Please state what role the funders took in the study.  If the funders had no role, please state: "The funders had no role in study design, data collection and analysis, decision to publish, or preparation of the manuscript.

4. Thank you for stating the following in the Acknowledgments Section of your manuscript: "This work was supported by the Partnership for Canada-Caribbean Climate Change Adaptation (ParCA) (http://parca.uwaterloo.ca/). The grant number is 106372-006. In addition, we acknowledge the logistics and supports provided by several organizations in Jamaica, including NEPA, NEPT, and CaribSave. Specially thanks go to Ms. Jacqueline daCosta, the former president of the Commonwealth Association of Planners and the Jamaica Institute of Planners. Our deepest gratitude goes to all individuals across Jamaica, participated in the different forms of the field survey"

Please remove any funding-related text from the manuscript and let us know how you would like to update your Funding Statement. Currently, your Funding Statement reads as follows: "This work was supported by the Partnership for Canada-Caribbean Climate Change Adaptation (ParCA) (http://parca.uwaterloo.ca/). The grant number is:  106372-006"

7. Please ensure that you include a title page within your main document. We do appreciate that you have a title page document uploaded as a separate file, however, as per our author guidelines (http://journals.plos.org/plosone/s/submission-guidelines#loc-title-page) we do require this to be part of the manuscript file itself and not uploaded separately.

8. Please include your full ethics statement in the ‘Methods’ section of your manuscript file. In your statement, please include the full name of the IRB or ethics committee who approved or waived your study, as well as whether or not you obtained informed written or verbal consent. If consent was waived for your study, please include this information in your statement as well. 

9. We note that Figure 2 in your submission contain map images which may be copyrighted. All PLOS content is published under the Creative Commons Attribution License (CC BY 4.0), which means that the manuscript, images, and Supporting Information files will be freely available online, and any third party is permitted to access, download, copy, distribute, and use these materials in any way, even commercially, with proper attribution. For these reasons, we cannot publish previously copyrighted maps or satellite images created using proprietary data, such as Google software (Google Maps, Street View, and Earth). For more information, see our copyright guidelines: http://journals.plos.org/plosone/s/licenses-and-copyright.

10. Please upload a new copy of Figure 2 as the detail is not clear. Please follow the link for more information: https://blogs.plos.org/plos/2019/06/looking-good-tips-for-creating-your-plos-figures-graphics/" https://blogs.plos.org/plos/2019/06/looking-good-tips-for-creating-your-plos-figures-graphics/

Reviewers' comments:

Reviewer's Responses to Questions

**Comments to the Author**

1. Is the manuscript technically sound, and do the data support the conclusions?

Reviewer #1: Partly

Reviewer #2: Partly

Reviewer #3: Partly

2. Has the statistical analysis been performed appropriately and rigorously? 

Reviewer #1: Yes

Reviewer #2: N/A

Reviewer #3: N/A

3. Have the authors made all data underlying the findings in their manuscript fully available?

Reviewer #1: No

Reviewer #2: No

Reviewer #3: No

4. Is the manuscript presented in an intelligible fashion and written in standard English?

Reviewer #1: No

Reviewer #2: No

Reviewer #3: Yes

5. Review Comments to the Author

Reviewer #1: General feedback:

The topic is very interesting and timely and developing methods for assessing resilience to climate change is much needed. However, I am recommending major revisions as there are several major aspects of the paper that need revision. My first major comment is that the writing style can be clunky and difficult to follow at times. For example, the author(s) use long paragraphs, long sentences, commas, and semi colons, instead of shorter paragraphs and sentences to break apart separate concepts and ideas. I suggest the authors thoroughly read through and revise the manuscript to address this issue and also try to be more concise throughout. Second, for readers without an urban design or planning background, the author(s) need to better define and explain some of the terminology. This manuscript should be more accessible to a general audience. Third, in the introduction and background sections, as well as the discussion section, the writing is very much kept at the theoretical level. I think it would be improved if specific examples, contexts, and case studies were integrated into these sections. It would make the ideas and arguments more concrete. As well as compare the Jamaica case with other similar cases. Fourth, the entire manuscript should be revised to be more concise and direct. At times the text strays or appears rambling instead of staying on point and being direct. Lastly, I suggest that the author(s) make an integrate to better integrate the various sources of data. In the results section, they mostly present the interview data, and while that it interested, it would be nicely complemented by some of the secondary data including maps.

Specific comments:

Abstract

• The use of the term ‘urban form’ is awkward in the abstract without any explanation or description

Introduction

• First paragraph, page 2 – this paragraph is long and describes multiple concepts. Try to break up the paragraph into digestible chunks. For example, start a new paragraph when the author(s) start explaining adaptation.

• Pg 4 – the paper objectives need to be more clearly stated. Perhaps by breaking up the objectives into an overarching objective or research question followed by sub-questions. Do not bury this within a longer paragraph, make it stand out to highlight them. As of now they are spread throughout two paragraphs and not concisely stated.

• Please define Urban morphology more clearly for those who are unfamiliar.

• In general, I would shorten this section and put much of this text into the following theoretical background sections. The introduction could then be reduced to a short opening paragraph that states the problem, a second short paragraph highlighting research gaps, and then a third short paragraph that clearly and concisely stats the paper/project objectives. Then the background can be provided in the next sections.

Contemporary discourses on urban resilience

• It is important to consider how resilience varies from adaptive capacity. They are often used interchangeably – so how do they differ when it comes to urban environments?

2.1 Urban form’s resilience and its measures

• A discussion of psychological resilience could be incorporated here when discusses Vale’s definition of urban resilience. While it developed separately from social-ecological resilience, there are certainly useful overlaps.

• Please define “urban form”

• Figure 1 is great – please refer to it more directly in the text.

• For each of the components in Figure 1 - please provide not only the theoretical explanations, but more concrete, real-world examples.

2.2 A framework for Urban morphological resilience

• Figure 1 does not belong here. It belongs in the previous section where it is explained and mentioned first. The same is true in Table 1. I suggest moving it earlier.

• Table 1 could also use some reformatting. Try to make the columns with a lot of text wider.

• This section is generally not needed. I would add it to the section above.

Case Study and Methods

3.1 Geographic focus

• Statistics about the impacts would be helpful, such as measured and/or projected sea level rise/

• There are virtually no references in this section, please do cite sources.

• There is also no explanation for why this area is a good place to implement the proposed framework.

3.2 Data sources

• Subtitle 3.2.2 is not formatted correctly

• I would suggest moving 3.2.3 Study limitations to the end of the methods section. That is where it is most typically found

3.3 Data management and analysis

4. Measuring Long Bay’s resilience

• I am not sure the term ‘measuring’ is accurate. The following discussions do not make judgement statements or overall measurements, instead they discuss various aspects that relate to resilience.

• The figures are helpful, but hard to interpret at times

• It also might be useful to more successfully integrate the secondary data and GIS data. Showing maps of current infrastructure systems would be helpful and nicely complement the quotes.

5. Discussion and conclusions

• I suggest moving the quote that opens up Section 5 to the very beginning of the entire manuscript as it helps situate the research.

• The first paragraph is just one long run-on sentence. Please do revise. Also a figure that summarizes the main findings would be incredibly helpful.

• The author(s) could more closely tie the discussion/conclusions back to the topics discussed in the introduction.

• The authors should compare their findings to similar studies in other geographical areas.

• Figure 8 belongs in the results section since it is a result.

Reviewer #2: The paper is focused in a very important issue as climate resilience is. The author proposes a framework for asses this climate resilience based, essentially, on urban design. Although the six points discussed in the framework are important, I think that, obviously, they are not enough to face all the problems related to climate change. For this reason, I think that a clearer discussion about the scope will be welcomed.

The literature review is wide but as the paper is mixing different areas, the final position remains unclear. In other words, is there other frameworks for assessing climate resilience? Could be better to focus on frameworks or in urban design?. I would appreciate a discussion on specific issue of the paper more than a general discussion on general models. In my case, the final idea of the discussion remains unclear. As the author said, I was expecting to limit the intersection between urban design and resilience, and to give specific actions for being done.

In fact, the sixfold framework comes from six urban design concepts, but it is not enough discussed if is the main paper contribution. The emergence of the six concepts should be better explained and to specify why these six concepts.

Having frameworks helps to give order to assessing methods but, in general, the application of indicators and measures is the path. In this case, the author present indicators and variables but it is not possible to apply them to case study.

Perhaps, storytelling needs to be rethought and strengthening the links between the framework and the application. In any case, the work done with all the interviews is very valuable and should be appreciated by the politicians and public servants of the case study.

I was expecting new ideas and new ways to adapt to climate change in the case of increasing sea-level and it was not the case. In fact, economic and political barriers appear, as it is usual.

Results are very interesting for decision-making but the author is proposing actions that should be part of a resilience plan.

For this reason, I consider that although the paper is interesting, it is unsuitable for publication in its present form. Nevertheless, the study shows sufficient potential to encourage the author to resubmit a revised version focusing on the main contribution to the academic community. In other case, the paper is nearer to a policy practice than an academic paper.

Reviewer #3: The authors are presenting an interesting work; however, the current version of the manuscript has many flaws. Introduction (section 1) and Literature Review (section 2) are unnecessarily long. These sections need to be combined and reduced. Especially the concepts explained in the section 2 need to be summarized and reduced. No need to explain in detail when they have a summary table (Table 1) which presents the same information. I prefer to rename the section 3 as Materials and Methods, and it need to be section 2. Section 4 should be Results and Discussion and need to be changed to section 3. I also wonder why they do not have a standalone conclusion section. Therefore, I recommend major revisions to this paper so that the authors can have an opportunity to greatly improve their work that can be useful and readable.

6. PLOS authors have the option to publish the peer review history of their article (what does this mean?). If published, this will include your full peer review and any attached files.

Reviewer #1: No

Reviewer #2: No

Reviewer #3: No

---

## [Author Response · Author response to Decision Letter 0]

22 Feb 2023

Reviewer 1

The topic is very interesting and timely and developing methods for assessing resilience to climate change is much needed. However, I am recommending major revisions as there are several major aspects of the paper that need revision. My first major comment is that the writing style can be clunky and difficult to follow at times. For example, the author(s) use long paragraphs, long sentences, commas, and semi colons, instead of shorter paragraphs and sentences to break apart separate concepts and ideas. I suggest the authors thoroughly read through and revise the manuscript to address this issue and also try to be more concise throughout. 

Authors’ response: 

We thank Reviewer 1 for the constructive and detailed comments and for appreciating our efforts. As advised, we have revised the entire manuscript and fixed the writing issue. 

Second, for readers without an urban design or planning background, the author(s) need to better define and explain some of the terminology. This manuscript should be more accessible to a general audience. 

Authors’ response: 

we have defined the key terminologies, i.e., urban form and morphology, in the texts and In the endnote.

Third, in the introduction and background sections, as well as the discussion section, the writing is very much kept at the theoretical level. I think it would be improved if specific examples, contexts, and case studies were integrated into these sections. It would make the ideas and arguments more concrete. As well as compare the Jamaica case with other similar cases. 

Authors’ response: 

We thank the reviewer for this critical observation. We have added a few examples (marked with Cyan) to the sections to relate to our theoretical contents. We also added a few cases to the discussion section to compare our findings from Jamaica. 

Fourth, the entire manuscript should be revised to be more concise and direct. At times the text strays or appears rambling instead of staying on point and being direct. 

Lastly, I suggest that the author(s) integrate to better integrate the various data sources. In the results section, they mostly present the interview data, and while that it interested, it would be nicely complemented by some of the secondary data, including maps.

Authors’ response: 

We agree with Reviewer 1’s comments. We have made the entire manuscript more concise. In the revised discussion section, we have included more information (from secondary data and other sources) to complement our findings. 

Specific comments:

Abstract

• The use of the term ‘urban form’ is awkward in the abstract without any explanation or description

Authors’ response: we have added a brief dentition of urban form in the abstract. 

Introduction

• First paragraph, page 2 – this paragraph is long and describes multiple concepts. Try to break up the paragraph into digestible chunks. For example, start a new paragraph when the author(s) start explaining adaptation.

• Pg 4 – the paper objectives need to be more clearly stated. Perhaps by breaking up the objectives into an overarching objective or research question followed by sub-questions. Do not bury this within a longer paragraph, make it stand out to highlight them. As of now they are spread throughout two paragraphs and not concisely stated.

• Please define Urban morphology more clearly for those who are unfamiliar.

• In general, I would shorten this section and put much of this text into the following theoretical background sections. The introduction could then be reduced to a short opening paragraph that states the problem, a second short paragraph highlighting research gaps, and then a third short paragraph that clearly and concisely stats the paper/project objectives. Then the background can be provided in the next sections.

Authors’ response: Thank you for this suggestion. We have shortened the section and made it concise. We have briefly added a brief definition of urban form and morphology to the main texts. Also, their detailed definition is available in the footnotes. The second last paragraph now discusses the paper’s main objective and key questions. 

Contemporary discourses on urban resilience

• It is important to consider how resilience varies from adaptive capacity. They are often used interchangeably – so how do they differ when it comes to urban environments?

Authors’ response: Thank you. We agree with the reviewers. The revised section explains the relationship between resilience and adaptive capacity in the context of urban environments. 

2.1 Urban form’s resilience and its measures

• A discussion of psychological resilience could be incorporated here when discusses Vale’s definition of urban resilience. While it developed separately from social-ecological resilience, there are certainly useful overlaps.

• Please define “urban form”

• Figure 1 is great – please refer to it more directly in the text.

• For each of the components in Figure 1 - please provide not only the theoretical explanations, but more concrete, real-world examples.

Authors’ response: We thank the Reviewer for pointing out this. In fact, urban morphology (i.e., the physical dimension of urban design) is a key component of our work. We didn’t pay much attention to “psychological resilience” because it remains out of our scope. Yet, as advised by the Reviewer, we have discussed psychological resilience briefly to help our readers become aware of it and understand a broader definition of resilience. 

We are glad that the reviewer finds the figure useful. We have added a few lines to explain it more. A definition of urban form is now available in the texts and endnotes. 

2.2 A framework for Urban morphological resilience

• Figure 1 does not belong here. It belongs in the previous section where it is explained and mentioned first. The same is true in Table 1. I suggest moving it earlier.

• Table 1 could also use some reformatting. Try to make the columns with a lot of text wider.

• This section is generally not needed. I would add it to the section above.

Authors’ response: 

We thank the reviewer for this suggestion. We have relocated Figure 1 and Table 1 and also shortened the table’s contents as much as possible since we discussed them in Section 2.1. Lastly, as advised by the reviewer, we have merged this section with the section above. 

Case Study and Methods

3.1 Geographic focus

• Statistics about the impacts would be helpful, such as measured and/or projected sea level rise/

• There are virtually no references in this section, please do cite sources.

• There is also no explanation for why this area is a good place to implement the proposed framework.

Authors’ response: 

We agree with the reviewer and appreciate pointing out this lack. We revised the section according to the suggestions and cited more sources. We revised Fig 2 (location map) to include more data. Also, the section includes more evidence and statistics. We also discussed a few reasons for selecting Negril as a good case/place (please see the last paragraph of Section 3.1) 

3.2 Data sources

• Subtitle 3.2.2 is not formatted correctly

• I would suggest moving 3.2.3 Study limitations to the end of the methods section. That is where it is most typically found

Authors’ response: 

We revised the section according to the reviewer’s suggestion. 

4. Measuring Long Bay’s resilience

• I am not sure the term ‘measuring’ is accurate. The following discussions do not make judgement statements or overall measurements, instead they discuss various aspects that relate to resilience.

• The figures are helpful, but hard to interpret at times

• It also might be useful to more successfully integrate the secondary data and GIS data. Showing maps of current infrastructure systems would be helpful and nicely complement the quotes.

Authors’ response: 

We have added more explanations based on secondary data and other evidence to complement the quotes. We agree that Figures (4-7) are a bit abstract. We kept them generic so that readers could understand the main themes (without the contextual complexity of Negril). 

5. Discussion and conclusions

• I suggest moving the quote that opens up Section 5 to the very beginning of the entire manuscript as it helps situate the research.

• The first paragraph is just one long run-on sentence. Please do revise. Also a figure that summarizes the main findings would be incredibly helpful.

• The author(s) could more closely tie the discussion/conclusions back to the topics discussed in the introduction.

• The authors should compare their findings to similar studies in other geographical areas.

• Figure 8 belongs in the results section since it is a result.

Authors’ response: 

We thank Reviewer 1 for this suggestion. We agree with it and move the quote to the beginning of the manuscript. We revised this section and long sentences. In fact, Figure 8 presents a summary of our assessment. We added a few similar contexts (e.g., in India and USA) to the section so as to compare and complement our findings. 

Reviewer 2

The paper is focused in a very important issue as climate resilience is. The author proposes a framework for asses this climate resilience based, essentially, on urban design. Although the six points discussed in the framework are important, I think that, obviously, they are not enough to face all the problems related to climate change. For this reason, I think that a clearer discussion about the scope will be welcomed.

Authors’ response: 

We thank Reviewer 2 for appreciating our attempts. We have revised the introduction to make its scope and objective clear. 

The literature review is wide but as the paper is mixing different areas, the final position remains unclear. In other words, is there other frameworks for assessing climate resilience? Could be better to focus on frameworks or in urban design?. I would appreciate a discussion on specific issue of the paper more than a general discussion on general models. In my case, the final idea of the discussion remains unclear. As the author said, I was expecting to limit the intersection between urban design and resilience, and to give specific actions for being done.

Authors’ response: 

We agree with Reviewer 2. We revised the literature review section and deleted some irrelevant points to clarify the main arguments. The revised manuscript now includes a few frameworks (for example, Allen 2001, Jabareen 2015, Quigley 2018, and Pickett’s 2013) that are also relevant to urban design but not directly associated with climatic variability. The revised sections also draw the theoretical foundation and application guidelines to integrate urban design and urban resilience in reference to other applications. We hope it makes the entire discussion clear, focused, and evidential. 

In fact, the sixfold framework comes from six urban design concepts, but it is not enough discussed if is the main paper contribution. The emergence of the six concepts should be better explained and to specify why these six concepts.

Authors’ response: 

We thank Reviewer 2 for pointing out these issues. We have tried to develop the framework as the main contribution of this paper. There are several concepts that have been used in the urban design/planning literature and practiced over decades. However, we found that only six of them have the potential to link urban design and climate resilience. In the revised section, we have explained it. 

Having frameworks helps to give order to assessing methods but, in general, the application of indicators and measures is the path. In this case, the author present indicators and variables but it is not possible to apply them to case study. Perhaps, storytelling needs to be rethought and strengthening the links between the framework and the application. 

In any case, the work done with all the interviews is very valuable and should be appreciated by the politicians and public servants of the case study.

I was expecting new ideas and new ways to adapt to climate change in the case of increasing sea-level and it was not the case. In fact, economic and political barriers appear, as it is usual.

Authors’ response: 

We thank the reviewer for identifying this shortcoming. We agree with Reviewer 2. The framework and its components are conceptual. All of their indicators may not be useful/applicable directly to a single case. Like in Negril, we tried to apply them to understand the baseline resilience based on the physical planning and application policies. The secondary data and the expert opinions were used (from our interviews) to verify/validate our findings or vice versa. However, we focused more on the physical environment and urban design. Unfortunately, we didn’t include the voice of politicians in this study. We believe their comments are important and would have added new socio-political insight to this study. We consider it as a limitation of this research and discuss it in Section 3.4 accordingly. 

Results are very interesting for decision-making but the author is proposing actions that should be part of a resilience plan. For this reason, I consider that although the paper is interesting, it is unsuitable for publication in its present form. Nevertheless, the study shows sufficient potential to encourage the author to resubmit a revised version focusing on the main contribution to the academic community. In other case, the paper is nearer to a policy practice than an academic paper.

Authors’ response: 

We thank you for the critical insight. In the revised conclusion we have highlighted the explicit contribution (theoretical and practical) of the paper to the academic community. 

Reviewer 3: 

The authors are presenting an interesting work; however, the current version of the manuscript has many flaws. Introduction (section 1) and Literature Review (section 2) are unnecessarily long. These sections need to be combined and reduced. Especially the concepts explained in the section 2 need to be summarized and reduced. 

Authors’ response: 

We thank Reviewer 3 for finding our work interesting. We agree that both sections (introduction and literature review) are long. We have cut a few irrelevant concepts and made them shortened and as precise as possible. 

The concepts in Section 2 represent the key components that contribute to the framework we have proposed. We have cut some of them and tried to reduce the section. We have added examples and different scholarly views that are relevant to make our arguments clear and logical. 

No need to explain in detail when they have a summary table (Table 1) which presents the same information. 

Authors’ response: 

To avoid redundancy, we have shortened Table 1 as much as possible, which now presents only a summary of Section 2.1. 

I prefer to rename the section 3 as Materials and Methods, and it need to be section 2. 

Section 4 should be Results and Discussion and need to be changed to section 3. 

Authors’ response: 

We have revised them accordingly. 

I also wonder why they do not have a standalone conclusion section. Therefore, I recommend major revisions to this paper so that the authors can have an opportunity to greatly improve their work that can be useful and readable.

Authors’ response: 

We thank the reviewer for this suggestion. The revised manuscript now has a separate “standalone conclusion.” It discusses the paper’s contribution with more evidence and contexts similar to our case.

---

## [Decision Letter · Decision Letter 1]

20 Mar 2023

PONE-D-21-17612R1A sixfold urban design framework to assess climate resilience: generative transformation in Negril, JamaicaPLOS ONE

Dear Dr. Dhar,

Thank you for submitting your manuscript to PLOS ONE. After careful consideration, we feel that it has merit but does not fully meet PLOS ONE’s publication criteria as it currently stands. Therefore, we invite you to submit a revised version of the manuscript that addresses the points raised during the review process.

We look forward to receiving your revised manuscript.

Kind regards,

Md Nazirul Islam Sarker

Academic Editor

PLOS ONE

Additional Editor Comments (if provided):

The author is advised to address all comments point-by-point.

Reviewers' comments:

Reviewer's Responses to Questions

**Comments to the Author**

1. If the authors have adequately addressed your comments raised in a previous round of review and you feel that this manuscript is now acceptable for publication, you may indicate that here to bypass the “Comments to the Author” section, enter your conflict of interest statement in the “Confidential to Editor” section, and submit your "Accept" recommendation.

Reviewer #1: All comments have been addressed

Reviewer #2: (No Response)

Reviewer #3: All comments have been addressed

2. Is the manuscript technically sound, and do the data support the conclusions?

Reviewer #1: Yes

Reviewer #2: Partly

Reviewer #3: Yes

3. Has the statistical analysis been performed appropriately and rigorously? 

Reviewer #1: Yes

Reviewer #2: N/A

Reviewer #3: N/A

4. Have the authors made all data underlying the findings in their manuscript fully available?

Reviewer #1: Yes

Reviewer #2: (No Response)

Reviewer #3: No

5. Is the manuscript presented in an intelligible fashion and written in standard English?

Reviewer #1: Yes

Reviewer #2: No

Reviewer #3: Yes

6. Review Comments to the Author

Reviewer #1: Thank you for the thorough revision. This paper is much improved from the original draft. However, I still find that the Results and Discussion section needs major revisions. See my more detailed comments below.

• The clean version has some awkward spacing and punctuation – likely from the track changes revisions. Please do review the clean version to edit these small mistakes.

• Introduction – The first quote and paragraph are not well linked with the second paragraph. Try to write a transition sentence or something that links the two. Overall, the introduction is much improved. It is concise and well cited.

• Contemporary discourses on urban resilience – This section is also much improved. It is well organized, and provides a thorough definition of each concept.

• Methods – Clear and concise

• Results and Discussion – This section is still lacking much of a ‘discussion’. It serves mostly to summarize the results, but does not place it into the larger context nor compare results with other research. There are very few citations. Additionally, there is no overarching section where the author(s) directly speak to how the results answer their research questions. This is important to include. It is also important to generalize these results. How could this framework be used and applied elsewhere? How do the results compare with other similar studies? This type of discussion is lacking, and this section is mostly a results section.

• Conclusion – some of this could be moved to the results and discussion section as the overall summary of the results. I do like how the authors have organized the last paragraph by the theoretical, empirical and practice-wise contributions – but I would recommend breaking up each of these into its own stand alone paragraph. That would help each contribution stand out.

Reviewer #2: It is more a theoretical article than an applied one. For this reason, the data doesn't support all conclusions.

It is not so nice for reading , in some parts you need two read twice or three times for understanding the sense of the content.

The article has improved but section 2 needs to better clarify what is the final goal. It is difficult to understand for people that is not coming from urban side. Storytelling is not well designed and some gaps remain in this section.

The title of 2.1 is not linked to the text, I’m not able to find measures of urban form’s resilience. In fact, in table 1 you are using “concepts” word and no measures. Title is not corresponding to the content of this point. In case you are using measures devoted to ways of measurement, it is necessary to clarify because, for me, the word measures are measures for action to be taken by someone.

In page 10, you need to change 3.1 for 2.1 Geographic focus because it is under 2. Materials and Methods section.

Concerning 3.4 Study limitations and advantages, my point is it is about data or about the study. In this case, you need to move the section to the end.

Results seems to derive of the framework application but it remains confusing. It could be better to stress this link in order to give more sense to the proposed framework.

I’m not used to include new references in conclusions. The discussion has been made previously.

Reviewer #3: I thank the authors for substantially improving their manuscript adhering to the reviewers’ comments. However, I suggest authors to go through the entire manuscript and check for typos and minor spell/grammar mistakes. It is also better to double check all the citations and references. In some places, fonts were different as well. Title of the section 4 needs to be just Results and Discussion and the rest in the title can be removed. Conclusion section is very long. I would suggest removing the citations and referencing to the previous work in the conclusions so they can cut it down to 2-3 paragraphs. Conclusions should highlight the key findings of the study with limitations and recommendations for future work. Therefore, I recommend minor revisions to this paper.

7. PLOS authors have the option to publish the peer review history of their article (what does this mean?). If published, this will include your full peer review and any attached files.

Reviewer #1: No

Reviewer #2: No

Reviewer #3: No

<quillbot-extension-portal></quillbot-extension-portal>

---

## [Author Response · Author response to Decision Letter 1]

20 Apr 2023

Response to Reviewers (Round 2)

PONE-D-21-17612R1

Title: A sixfold urban design framework to assess climate resilience: generative transformation in Negril, Jamaica

Reviewer 1

“Thank you for the thorough revision. This paper is much improved from the original draft. However, I still find that the Results and Discussion section needs major revisions. See my more detailed comments below. The clean version has some awkward spacing and punctuation – likely from the track changes revisions. Please do review the clean version to edit these small mistakes.”

Authors’ response: 

We thank the respected reviewer for reviewing our manuscript again and for your valuable suggestions. We have carefully considered your suggestions to revise the manuscript thoroughly in this round and fix minor editorial issues (including typos, spacing and punctuation). 

“Introduction – The first quote and paragraph are not well linked with the second paragraph. Try to write a transition sentence or something that links the two.” 

Authors’ response: 

We agree with you. We have revised the first two paragraphs to improve the link cohesively. 

“Overall, the introduction is much improved. It is concise and well cited. 

Contemporary discourses on urban resilience – This section is also much improved. It is well organized, and provides a thorough definition of each concept.

Methods – Clear and concise.”

Authors’ response: 

Thank you so much for valuing our efforts. 

“Conclusion – some of this could be moved to the results and discussion section as the overall summary of the results. 

Authors’ response: 

We really appreciate your suggestion. Accordingly, we revised the Results and Discussion (the last part) and Conclusions section significantly. 

"I do like how the authors have organized the last paragraph by the theoretical, empirical and practice- wise contributions – but I would recommend breaking up each of these into its own stand alone paragraph. That would help each contribution stand out.” 

Authors’ response: 

We are glad to know that our discussion on theoretical, empirical and practice-wise contributions has become effective. In this revised version, we have separated each contribution. 

Reviewer 2

“It is more a theoretical article than an applied one. For this reason, the data doesn't support all conclusions. It is not so nice for reading, in some parts, you need two read twice or three times for understanding the sense of the content. 

The article has improved but section 2 needs to better clarify what is the final goal. It is difficult to understand for people that is not coming from urban side. Storytelling is not well designed and some gaps remain in this section.”

Authors’ response: 

We thank the respected reviewer for highlighting the key notion of our paper, i.e., toward establishing the theoretical link among interdisciplinary components in responding to climatic risks. 

We revised our manuscript thoroughly to improve its readability. In doing so, we have shortened sentences to simplify our arguments in many sections. We also revised Section 2 (i.e., Contemporary discourses on urban resilience), where we developed the foundation of our theoretical framework – one of the major goals of this paper (as discussed on Page 4 and in our abstract). 

We consider your comments seriously. In this revision, we have revised all sections to be suitable for a general audience, particularly those who are not from the urban side. We defined “urban design” in Introduction for such readers. Also, a new endnote (which describes urban design more along with its origin and scope) has been added for general readers who are less/not familiar with urban design but are interested in the field. We believe that the three endnotes (on urban design, urban form, and urban morphology) will help such readers. 

“The title of 2.1 is not linked to the text, I’m not able to find measures of urban form’s resilience. In fact, in table 1 you are using “concepts” word and no measures. Title is not corresponding to the content of this point. In case you are using measures devoted to ways of measurement, it is necessary to clarify because, for me, the word measures are measures for action to be taken by someone.”

Authors’ response: 

We thank you for pointing this out. We completely agree with you. We changed the title of Section 2.1 to better link to the contents of the section as well as Table 1. 

“In page 10, you need to change 3.1 for 2.1 Geographic focus because it is under 2. Materials and Methods section.”

Authors’ response: 

We have corrected all numbers of the titles and subtitles and their sequences. 

“Concerning 3.4 Study limitations and advantages, my point is it is about data or about the study. In this case, you need to move the section to the end.”

Authors’ response: 

We thank the reviewer for sharing the critical insight. We have moved this section to conclusions.

“Results seems to derive of the framework application but it remains confusing. It could be better to stress this link in order to give more sense to the proposed framework.”

Authors’ response: 

We thank the reviewer for this suggestion. Throughout the manuscript, we referred to many cases and applications of other areas to help our readers understand the link. We revised our conclusions and added a new Section 4.7 (Negril’s resilience and a path toward generative transformation) to clarify the link (both theoretical and empirical) that is the key objective of the proposed framework. 

“I’m not used to include new references in conclusions. The discussion has been made previously.” 

Authors’ response: 

In this revision, we removed all references from the conclusions. 

Reviewer 3: 

“I thank the authors for substantially improving their manuscript adhering to the reviewers’ comments. However, I suggest authors to go through the entire manuscript and check for typos and minor spell/grammar mistakes. It is also better to double check all the citations and references. In some places, fonts were different as well.” 

Authors’ response: 

We thank the respected reviewer for considering our last revision effective and improved. At this round, we have double-checked and addressed all minor editorial mistakes and citation issues to improve the quality of our paper further. 

“Title of the section 4 needs to be just Results and Discussion and the rest in the title can be removed.” 

Authors’ response: 

We revised the titles as advised. Thank you. 

“Conclusion section is very long. I would suggest removing the citations and referencing to the previous work in the conclusions so they can cut it down to 2-3 paragraphs. Conclusions should highlight the key findings of the study with limitations and recommendations for future work.” 

Authors’ response: 

Thank you for this intellectual advice. We have revised our conclusions significantly. We condensed the section, highlighting our key findings and contribution with limitations and recommendations. We have also removed all citations.

---

## [Decision Letter · Decision Letter 2]

17 May 2023

PONE-D-21-17612R2A sixfold urban design framework to assess climate resilience: generative transformation in Negril, JamaicaPLOS ONE

Dear Dr. Dhar,

Thank you for submitting your manuscript to PLOS ONE. After careful consideration, we feel that it has merit but does not fully meet PLOS ONE’s publication criteria as it currently stands. Therefore, we invite you to submit a revised version of the manuscript that addresses the points raised during the review process.

We look forward to receiving your revised manuscript.

Kind regards,

Dr. Md Nazirul Islam Sarker

Academic Editor

PLOS ONE

Journal Requirements:

Additional Editor Comments:

The author is advised to address the further comments of the reviewer 1 point-by-point.

Reviewers' comments:

Reviewer's Responses to Questions

**Comments to the Author**

1. If the authors have adequately addressed your comments raised in a previous round of review and you feel that this manuscript is now acceptable for publication, you may indicate that here to bypass the “Comments to the Author” section, enter your conflict of interest statement in the “Confidential to Editor” section, and submit your "Accept" recommendation.

Reviewer #1: (No Response)

Reviewer #3: All comments have been addressed

2. Is the manuscript technically sound, and do the data support the conclusions?

Reviewer #1: Yes

Reviewer #3: Yes

3. Has the statistical analysis been performed appropriately and rigorously? 

Reviewer #1: Yes

Reviewer #3: Yes

4. Have the authors made all data underlying the findings in their manuscript fully available?

Reviewer #1: Yes

Reviewer #3: Yes

5. Is the manuscript presented in an intelligible fashion and written in standard English?

Reviewer #1: Yes

Reviewer #3: Yes

6. Review Comments to the Author

Reviewer #1: The authors have not addressed all of my revisions and concerns. They did not respond to the following and important suggestion from the last round of revisions:

"Results and Discussion – This section is still lacking much of a ‘discussion’. It serves mostly to summarize the results, but does not place it into the larger context nor compare results with other research. There are very few citations. Additionally, there is no overarching section where the author(s) directly speak to how the results answer their research questions. This is important to include. It is also important to generalize these results. How could this framework be used and applied elsewhere? How do the results compare with other similar studies? This type of discussion is lacking, and this section is mostly a results section."

They did add one section, but the rest of the section was not revised according to my suggestions.

Reviewer #3: (No Response)

7. PLOS authors have the option to publish the peer review history of their article (what does this mean?). If published, this will include your full peer review and any attached files.

Reviewer #1: No

Reviewer #3: No

---

## [Author Response · Author response to Decision Letter 2]

23 May 2023

Reviewer 1

The authors have not addressed all of my revisions and concerns. They did not respond to the following and important suggestion from the last round of revisions:

"Results and Discussion – This section is still lacking much of a ‘discussion’. It serves mostly to summarize the results, but does not place it into the larger context nor compare results with other research. There are very few citations. Additionally, there is no overarching section where the author(s) directly speak to how the results answer their research questions. This is important to include. It is also important to generalize these results. How could this framework be used and applied elsewhere? How do the results compare with other similar studies? This type of discussion is lacking, and this section is mostly a results section."

They did add one section, but the rest of the section was not revised according to my suggestions.

Authors’ response: 

We considered (and are considering) all reviewers’ comments seriously to take every opportunity to improve our manuscript. In fact, we addressed the concern of the respected Reviewer-1 and revised our manuscript accordingly. We also wrote how we made the revisions in our response letter. However, unfortunately, our responses to this particular section have somehow been missed/deleted while handling the file. It was a mistake, but it happened unintentionally. Please accept our sincere apologies.

I would like to add the following responses that were missed in our previous letter: 

“In this revision, based on our data, we discussed our findings in Sections 4.1 to 4.6. Each of these sections assesses the study area based on our proposed six concepts while connecting the field of urban design (and planning) and climate change resilience. Some of our concepts have long been discussed in the literature. For example, Ecological Sensitivity has existed in “urban ecology,” “design with nature,” and “landscape ecological urbanism” for a while, and their renewed interest in “eco-based adaptations” and “nature-based solutions” is emerging given the climate crisis. However, other concepts are new to the field (e.g., Redundancy and Indeterminacy), and their applications are rare in the context of climate change. Section 2.1 discusses it and presents a review of international cases (albeit they don’t relate to climate change) to clarify the six concepts and their similar applications worldwide. We didn’t repeat them in Sections 4.1 to 4.6. Instead, these sections shed more light on the application of each concept to the Long Bay area so as to understand and assess its resilience through its spatial layout and physical configuration–an objective of this paper. To generalize our results, we have added a new section (Section 4.7: Negril’s resilience and a path toward generative transformation) to explain our results in a broader context while comparing them with other studies.” 

In this round, we thank the respected reviewer again for reading and sharing his/her/their comments with us. We take another opportunity to carefully review the newly added Section 4.7 (marked in yellow) and revise the entire section again. The section now includes a few examples from the Caribbean and similar contexts to generalize and compare our results. It helps readers understand our concepts better and their applications beyond Negril. We believe the contents of this section also help readers understand the paper’s contribution (theoretical, empirical, and practice-wise), as we discussed in the following Conclusions sections.

---

## [Decision Letter · Decision Letter 3]

5 Jun 2023

A sixfold urban design framework to assess climate resilience: generative transformation in Negril, Jamaica

PONE-D-21-17612R3

Dear Dr. Dhar,

We’re pleased to inform you that your manuscript has been judged scientifically suitable for publication and will be formally accepted for publication once it meets all outstanding technical requirements.

Kind regards,

Md Nazirul Islam Sarker, PhD

Academic Editor

PLOS ONE

Additional Editor Comments (optional):

Congratulations! I sincerely appreciate your patience and hard work in addressing the extensive queries of the editor and three reviewers. Kindly stay in touch with the production team for the remaining publication process.

Reviewers' comments:

Reviewer's Responses to Questions

**Comments to the Author**

1. If the authors have adequately addressed your comments raised in a previous round of review and you feel that this manuscript is now acceptable for publication, you may indicate that here to bypass the “Comments to the Author” section, enter your conflict of interest statement in the “Confidential to Editor” section, and submit your "Accept" recommendation.

Reviewer #1: All comments have been addressed

2. Is the manuscript technically sound, and do the data support the conclusions?

Reviewer #1: Yes

3. Has the statistical analysis been performed appropriately and rigorously? 

Reviewer #1: Yes

4. Have the authors made all data underlying the findings in their manuscript fully available?

Reviewer #1: Yes

5. Is the manuscript presented in an intelligible fashion and written in standard English?

Reviewer #1: Yes

6. Review Comments to the Author

Reviewer #1: Thank you for the thoughtful response. You have thoroughly addressed my feedback.

7. PLOS authors have the option to publish the peer review history of their article (what does this mean?). If published, this will include your full peer review and any attached files.

Reviewer #1: No

---

## [Editor Report · Acceptance letter]

15 Jun 2023

PONE-D-21-17612R3 

A sixfold urban design framework to assess climate resilience: generative transformation in Negril, Jamaica 

Dear Dr. Dhar:

I'm pleased to inform you that your manuscript has been deemed suitable for publication in PLOS ONE. Congratulations! Your manuscript is now with our production department. 

Kind regards, 

on behalf of

Dr. Md Nazirul Islam Sarker 

Academic Editor

PLOS ONE